# A satellite-based analysis of semi-direct effects of biomass burning aerosols on fog and low cloud dissipation in the Namib Desert

Alexandre Mass[1,2], Hendrik Andersen[1,2], Jan Cermak[1,2], Paola Formenti[3], Eva Pauli[1,2], and Julian Quinting[1]

[1]Karlsruhe Institute of Technology (KIT), Institute of Meteorology and Climate Research, Karlsruhe, Germany
[2]Karlsruhe Institute of Technology (KIT), Institute of Photogrammetry and Remote Sensing, Karlsruhe, Germany
[3]Université Paris Cité and Univ Paris Est Creteil, CNRS, LISA, F-75013 Paris, France

**Correspondence:** Alexandre Mass (alexandre.mass@kit.edu)

**Abstract.** In the Namib Desert fog is the only regular water input and thus a crucial water source for its fauna and flora. Each year between June and October absorbing biomass burning aerosols (BBA) are overlying the stratocumulus clouds in the adjacent Southeast Atlantic. In some synoptic settings, this layer of BBA reaches Namibia and its desert, where it interacts with coastal fog and low clouds (FLCs). In this study, a novel 15-year data set of geostationary satellite observations of FLC dissipation time in the Namib Desert is used, along with reanalysis data, to better understand the potential semi-direct effects of BBA on FLC dissipation in the Namib, i.e., through adjustments of atmospheric stability and thermodynamics via the interaction of aerosols with radiation. This is done by investigating both the time of day when FLCs are dissolving and synoptics depending on BBA loading. It is found that FLC dissipation time is significantly later on high BBA loading days. BBA are transported to the Namib along moist free-tropospheric air by a large-scale anticyclonic recirculation pattern. At the surface, the associated longwave heating strengthens a continental heat low, which modifies the circulation and boundary layer moisture along the coastline, complicating the attribution of BBA effects. During high BBA days, the vertical profiles of the temporal development of air temperatures highlight contrasting day and nighttime processes modifying the local inversion. These processes are thought to be driven by greenhouse warming by the moisture in the BBA plumes and BBA absorption (only during daytime). A statistical learning framework is used to quantify meteorological and BBA influences on FLC dissipation time. The statistical model is able to reproduce the observed differences in FLC dissipation time between high and low BBA days and attributes these differences mainly to differences in circulation, boundary layer moisture and near-surface air temperature along the coastline. However, the model is underfitting and is not able to reproduce the majority of the FLC dissipation variability. While the model does not suggest that BBA patterns are important for FLC dissipation, the findings show how the moist BBA plumes modify local thermodynamics to which FLC dissipation is shown to be sensitive. The findings highlight the challenges of disentangling meteorological and aerosol effects on cloud development using observations and invite detailed modeling analyses of the underlying processes, for example, with large eddy simulations.

# 1 Introduction

Fog is the most relevant water source in the hyperarid Namib Desert and is the key to the survival of many species (Louw and Holm, 1972; Seely and Henschel, 1998; Ebner et al., 2011; Warren-Rhodes et al., 2013; Gottlieb et al., 2019; Wang et al., 2019). Regional climate simulations of southern Africa suggest a warmer and drier climate in the future (James and Washington, 2013; Maúre et al., 2018), so that fog might become even more relevant as a water source for local ecosystems. However, weather and climate models struggle to adequately represent fog in general (Vautard et al., 2009; Price et al., 2018), and a lack of understanding of the processes involved in the formation, maintenance and dissipation of fog in the Namib limits robust projections of possible future developments of this system.

The diurnal coastal circulation in the region features sea breezes during the day as cooler ocean air moves inland and land breezes at night as cooler air flows from land to sea (Lindesay and Tyson, 1990), facilitating the advection of fog, which is thought to predominantly form in the stable marine boundary layer over the cool Benguela Current and is then advected inland. (Olivier and Stockton, 1989; Seely and Henschel, 1998; Cermak, 2012; Andersen et al., 2019, 2020; Spirig et al., 2019). Fog most frequently occurs along the coastline, but can extend more than 100 km inland (Olivier, 1995; Cermak, 2012; Andersen et al., 2019). The seasonal cycle of fog occurrence varies with latitude as well as with distance to the coast. The maximum fog occurrence in low-lying coastal locations is during austral winter whereas during austral summer primarily lifted stratus clouds are advected that manifest as fog only further inland where they are intercepted by the land (Lancaster, 1984; Seely and Henschel, 1998; Andersen et al., 2019). The day-to-day variability of fog and low clouds (FLCs) in central Namib has been found to be associated with distinct synoptic-scale patterns that determine the large-scale transport of free-tropospheric moisture, controlling fog occurrence in two main ways. 1) Dry free-tropospheric air is advected from the southwest over the marine coastal regions where FLCs typically form, facilitating a strong cloud-top radiative cooling that helps sustain the low cloud layer. 2) At the same time, tropical moist air is advected over continental southern Africa, causing surface heating and the development of a heat low which drives the inland advection of the cloudy marine air masses (Andersen et al., 2020). While these findings help understand links between synoptic-scale variability and fog occurrence, the understanding of Namibian fog processes is still limited. In particular, possible influences of atmospheric aerosol particles on the life cycle of fog and its properties have not been studied to date.

A seasonally recurrent feature of Central Africa is the large-scale burning of biomass that emits large amounts of biomass burning aerosols (BBA) into the atmosphere between June and October (Zuidema et al., 2016). Lifted into the free troposphere over the continent, these BBA are frequently transported over the southern Atlantic ocean by my mid-tropospheric easterly winds between 5° and 15°S (Adebiyi et al., 2015). As wet scavenging is negligible in this region, the BBA plumes frequently persist for a relatively long time above the marine boundary-layer clouds (Wilcox, 2010; Adebiyi and Zuidema, 2016). Recent international projects have focussed on BBA effects on low clouds over the Southeast Atlantic Ocean, in particular ORACLES (ObseRvations of Aerosols above CLouds and their intEractionS), SEALS (Sea Earth Atmosphere Linkages Study in southern Africa), AEROCLO-sA (Aerosol Radiation and Clouds in southern Africa), LASIC (Layered Atlantic Smoke Interactions with Clouds) and CLARIFY (Clouds and Aerosol Radiative Impacts and Forcing) (Zuidema et al., 2016; Formenti et al.,

2019; Haywood et al., 2021; Redemann et al., 2021). One of the overall findings of these projects and related research is the important role of BBA for marine low-cloud development via semi-direct and indirect effects (e.g., Fuchs et al., 2017; Adebiyi and Zuidema, 2018; Diamond et al., 2018; Gordon et al., 2018; Deaconu et al., 2019; Herbert et al., 2019; Zhang and Zuidema, 2019).

The absorbing BBA layer leads to local heating, mostly in the free troposphere, but also reduces the downward solar radiation received at the surface, leading to a cooling there. As a consequence, BBA exert a strong positive direct radiative effect and changes the local temperature profile (Deaconu et al., 2019; De Graaf et al., 2020). Recent studies have shown that BBA layers tend to increase the stability of the lower troposphere, leading to stronger, lower-lying inversions with less entrainment of free-tropospheric air into the boundary layer and thus an increased cloudiness and cloud thickness (Wilcox, 2010; Sakaeda

et al., 2011; Gordon et al., 2018; Deaconu et al., 2019). However, BBAs are frequently transported within an anomalously moist atmospheric layer (Adebiyi et al., 2015) that modifies the longwave cooling rate of surfaces below (Adebiyi et al., 2015; Adebiyi and Zuidema, 2018; Deaconu et al., 2019), thereby complicating the attribution of BBA effects. BBA-induced atmospheric heating can also lead to a reduction in cloudiness if the heating occurs within the cloud layer due to increased evaporation (Hansen et al., 1997). Most of the research in this field is focused on the marine environment of the Southeastern

Atlantic, where the majority of the BBA plumes are transported. However, under different weather systems, a substantial amount of BBA plumes circulate to the free troposphere of the Southwestern African coastline (Flamant et al., 2022), leading to high-aerosol optical depth (AOD) situations in the central Namib (Adebiyi and Zuidema, 2016; Adesina et al., 2019). Potential atmospheric effects, and especially semi-direct effects of BBA on Namib-region FLCs are highly likely in such situations. Semi-direct effects in this study refer to the large-scale semi-direct effects, as defined in Diamond et al. (2022), involving

atmospheric thermodynamic, stability, and circulation adjustments resulting from the absorption or scattering of solar radiation by aerosols. As Namib-region FLCs primarily occur during nighttime and typically dissipate shortly after sunrise (Andersen and Cermak, 2018; Andersen et al., 2019; Spirig et al., 2019), and semi-direct effects of the absorbing BBA are expected to be most pronounced during daytime, the strongest effects of BBA may be expected during the dissipation of FLCs.

The goals of this study are thus to better understand possible BBA semi-direct effects on the dissipation of FLCs in the

Namib and to attempt to disentangle the BBA effects from other meteorological covariates. To this end, a 15-year time series of geostationary satellite observations of FLCs in the Namib is analyzed together with reanalysis data to characterize situations under contrasting BBA loading, and used in a statistical learning framework to quantify and partially disentangle meteorological and BBA influences on FLC dissipation time. The guiding hypothesis is that during the biomass burning season, BBA plumes lead to a stronger, lower-lying inversion and slower early morning planetary boundary-layer development, resulting in

a longer FLC lifetime.

## 2   Data and methods

This study uses multiple data sets from different space-borne sensors, reanalysis products and statistical analysis methods to characterize the dissipation of FLCs and its potential links to BBA in the Namib region. As the study focuses on the interactions

between FLCs and BBA, all analyses are conducted during the BBA season from June to October (De Graaf et al., 2014) over
a 15-year period (2004–2015). The spatial domain of this study is the western coastline of southern Africa (5°–30°S and
0°–25°E). Two regions with a high frequency of FLC occurrence were delineated by Andersen and Cermak (2018) and are
the particular focus of this study: the Central Namib (CN; 22°–24°S) and the Angolan Namib (AN; 15°–17°S) (Fig. 1). Both
regions have a similar topography with a relative narrow coastal plain and a central plateau further inland. The two regions are
mostly flat and arid except in the northernmost part of AN which features low cliffs. The biggest difference lies in the distance
to the BBA emission sources, with AN being closer. Within each region, FLCs are advected inland and therefore their life cycle
characteristics (i.e. time of advection and dissipation) are dependent on the distance to the coastline (Andersen and Cermak,
2018; Andersen et al., 2019). To control for the influence of the coastal proximity on FLC life cycle characteristics, FLCs are
considered only within the first 25 km from the coastline in the coastal plain.

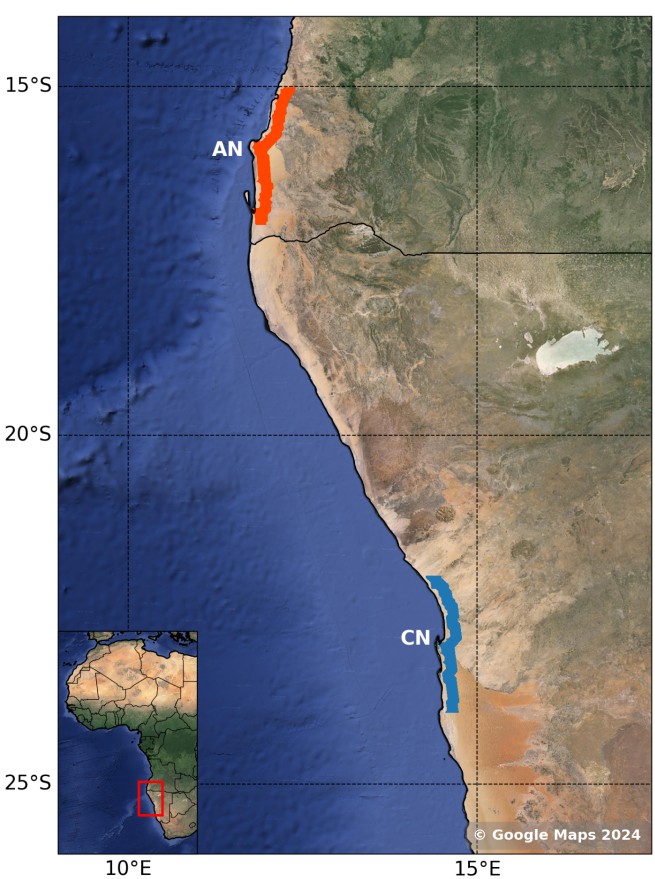

**Figure 1.** Overview of the study area, with the Angolan Namib region in orange and the Central Namib region in blue. The two regions are
defined by excluding the two first pixels after the coastline and within 25 km of it.

## 2.1 Satellite observations of FLC dissipation time

The data set of FLC dissipation time is created in two steps. First, the FLC detection algorithm developed by Andersen and Cermak (2018) is used on data from the Spinning Enhanced Visible and Infrared Imager (SEVIRI) sensor, mounted on the geostationary Meteosat Second Generation (MSG) satellites. SEVIRI provides spatiotemporally coherent observations of clouds. It features a spatial resolution of 3 km at nadir and a repeat rate of 15 min (Schmetz et al., 2002). The FLC detection algorithm is able to identify FLCs consistently at all times of the day by using infrared observations and a combination of threshold tests and image analysis techniques. Extensive validation against surface observations has shown a good performance (probability of detection of 94%, a false-alarm rate of 12% and an overall correctness of classification of 97%.). It should be noted that this satellite technique does not discriminate between fog and lifted stratus situations.

In the second step, a statistical method is applied to objectively detect the dissipation time of FLCs from the initial binary FLC data set produced in step 1. This method, as described by Pauli et al. (2022), employs logistic regression to predict the probability of a data point belonging to one of two binary classes: FLC or no FLC. By defining the transition between classes when this probability exceeds 50%, it becomes possible to determine the time of the transition from one class to the other and, as a result, determine the formation and dissipation times of each individual FLC event. A minor adjustment to the initial algorithm was implemented: the number of consecutive 15-minute time steps with the presence (or absence) of FLC to define an event is 4 instead of 10 because in Namibia the FLC events are more stable and the satellite view is less frequently interrupted by high cloud advection than in Europe where the method was originally developed.

The resulting data set provides the daily UTC time of FLC dissipation from 2004 to 2018, with a 15-minute temporal step (allowing for 96 possible dissipation times from 00:00 to 23:45) and a spatial resolution of 3 x 3 km. In both regions, dissipation begins shortly after sunrise, reaching a maximum at 8 UTC, which can be attributed to stronger solar irradiance (Andersen and Cermak, 2018). This is followed by a decrease until a daily minimum is observed around 12 UTC. It should be noted that the two pixels closest to the coastline are excluded in this analysis as they tend to be noisy, FLCs can hover along the coastline intermittently covering coastal locations, leading to uncertain estimation of dissipation time. Additionally, any absence of FLC events, such as during clear sky days, is discarded and treated as missing data.

## 2.2 CAMS global reanalysis

To quantify the atmospheric BBA load, the Copernicus Atmosphere Monitoring Service (CAMS) reanalysis data from the ECMWF are used. This product features a spatial resolution of 0.75° x 0.75° and a temporal resolution of 3 hours (Inness et al., 2019). While more observational-based products exist, such as aircraft measurements from field campaigns in the region, which are very useful in case studies, a reanalysis data set is preferred in this study to avoid issues related to missing data, cloud interference, and to facilitate combination with meteorological reanalysis data in a statistical framework. Additionally, the AOD data was extensively validated using observations in Gueymard and Yang (2020), where the authors found a small but existing tendency in CAMS to underestimate AOD across Africa. However, they conclude that for many applications, this data set offers significant advantages over customary observational-based products.

The black carbon aerosol optical depth at 550 nm (BCAOD) data are used as a proxy for the BBA loading as several aircraft campaigns have shown that southern African BBA fractional black carbon content is high in both number and mass of total particles (Taylor et al., 2020; Denjean et al., 2020). Half-day averages from 00:00 to 12:00 UTC are used to capture BBA that may influence the morning dissipation of FLCs, which is the most frequent dissipation time (Andersen and Cermak, 2018). To study possible BBA effects on FLC dissipation, two groups of days are created for each region: days with high BBA loading (BCAOD $\geq$ 75th percentile) and days with low BBA loading (BCAOD $\leq$ 25th percentile). They are referred to as 'high BBA days' and 'low BBA days,' each containing around 300 days after discarding those where at least one pixel of the region is outside the defined thresholds.

In the CN, the first quartile is characterized by a high density, followed by an almost exponential decrease into and throughout the third quartile. In contrast, the AN exhibits a more linear increase in density within the first quartile, followed by a gradual decline as BCAOD increases (see Appendix A). These differences are attributed to the distance from emission sources.

## 2.3 ERA5 reanalysis

Large-scale meteorological conditions are represented in the ERA5 reanalysis data from the European Centre for Medium-Range Weather Forecast (ECMWF). ERA5 features a 0.25° x 0.25° spatial and an hourly temporal resolution (Hersbach et al., 2020). To characterize large-scale dynamic and thermodynamic conditions, fields of mean sea level pressure (MSLP), geopotential height at 650 hPa (Z650), 2m air temperature (T2M), sea surface temperature (SST), specific humidity at 650 hPa (Q650) and 975 hPa (Q975) and u and v components of wind at 650 hPa and 10 meters above the surface are analyzed. Additionally, atmospheric temperature at all ERA5 pressure levels between 1000 and 500 hPa is used. Estimated inversion strength (EIS) is computed based on the method developed by Wood and Bretherton (2006). If not explicitly indicated, 8 UTC fields are selected to represent conditions during the typical time of FLC dissipation (Andersen and Cermak, 2018). Additionally, the high and low BBA days defined with CAMS are used to average ERA5 fields to obtain the meteorological situations on these days. However, aerosol effects are not explicitly represented in ERA5. The model is forced by climatological distributions of optical depth from sea salt, soil/dust, organic, black carbon and sulphate (Hersbach et al., 2015). Therefore, it is important to keep in mind that this prescribed forcing constrains the model's radiative environment by limiting the effects of aerosol variability. An additional limitation of the data is related to the assimilation scheme of ERA5, which uses 12-hour windows from 09 UTC to 21 UTC (Hersbach et al., 2020). In regions with sparse observations, such as the Namib Desert, the reanalysis heavily relies on satellite data and the underlying numerical weather model, which can result in discontinuities during these times.

## 2.4 CALIPSO

To characterize the cloud and aerosol layers, as well as their relative positions, data from the active-sensor platform of the Cloud-Aerosol Lidar and Infrared Pathfinder Satellite Observation (CALIPSO) is used and presented in Figure 2. CALIPSO is equipped with the Cloud-Aerosol Lidar with Orthogonal Polarization (CALIOP), which features a vertical resolution of 30 m and a horizontal resolution of 333 m (Vaughan et al., 2004). Here, the level 2 CALIPSO 5 km merged cloud and aerosols layer

product (version 4.20) is used for the period 13 June 2006–31 December 2018 from both nighttime and daytime overpasses. In contrast to all other data in this study, 13 June 2006 is the earliest available date because CALIPSO was launched in April 2006. The FLC layers were derived similarly to the approach presented in Cermak (2018). First, the cloud layer altitude was calculated by subtracting the terrain altitude from the observed feature altitude. Then all cloud layers with a cloud top height equal to or smaller than 2 km were defined as FLC. To characterize the vertical distribution of BBA, smoke layers were derived using the Feature Classification Flags product (Vaughan et al., 2023). It should be noted that for aerosol layers located below 1 km above the surface, the instrument cannot differentiate between smoke and urban pollution, but in this region, urban pollution is not expected to be relevant.

## 2.5 Ridge Regression

In this study, statistical modelling of FLC dissipation times is done by using meteorological fields from ERA5 and atmospheric BBA loading from CAMS as predictors. Typically, local meteorological fields are used as predictors for the statistical modelling of FLCs and Namibian stratocumulus clouds (e.g. Adebiyi and Zuidema, 2018; Fuchs et al., 2018; Zipfel et al., 2022). However, the Namibian FLC system is largely controlled by regional wind systems and their modulation through synoptic-scale variability (Andersen et al., 2020), and approximating such large-scale dynamics using local pressure or wind fields is difficult. Spatial neural networks would be an ideal tool for this; however, they require a large number of data points, and their interpretation and sensitivity estimation are more challenging. Therefore, ridge regression, a regularized linear model (James et al., 2021), is used to predict FLC occurrence based on large-scale spatial meteorological fields, as in Andersen et al. (2020). The regularization helps deal with the high number of correlated predictors which could lead to a high-variance (overfitting) problem if a classical statistical model was used. In ridge regression, the regularization is controlled $\lambda$ which shrinks the coefficients of the model towards zero using the L2 penalty: the squared magnitudes of the coefficient value are added as a penalty term to the loss function (Friedman et al., 2010). Usually, the optimal value of the tuning parameter $\lambda$ is defined using cross-validation, which derives a value for $\lambda$ which is high enough to reduce overfitting issues while simultaneously not impairing the predictive skill of the model.

Here, to reduce model variance and increase the robustness of model predictions, an ensemble of 50 ridge regression models is used. Each regression model is trained on randomly selected $80\%$ of the data, withholding $20\%$ for testing. To ensure comparability between the individual models, $\lambda$ is set to a fixed value of 10000, a stronger regularization than determining $\lambda$ through cross-validation (optimal values range from 3000 to 7000). However, analyses of the training/test results showed only a marginal loss of predictive skill while substantially reducing overfitting when setting $\lambda$ to 10000.

The ridge regression method is applied to predict FLC dissipation times for the days with high and low BBA derived from CAMS. The predictors are the spatial fields of MSLP, Z650, T2M, SST, Q650, Q975 and EIS (as defined in section 2.3) from ERA5 and BCAOD from CAMS. For the different predictors to be comparable, standardization of the features was done by removing the mean and scaling to unit variance. To harmonize the spatial resolution of all predictors, the ERA5 fields (0.25° x 0.25°) are upscaled to match the coarser resolution of CAMS (0.75° x 0.75°). However, the rescaling of the predictand is unnecessary because a spatial median of dissipation times is calculated for each region at each time step. Because FLC events

are highly dependent on the synoptic-scales meteorological processes in the region (Andersen et al., 2020), the spatial domain of the predictors is set to be centred on the zone of interest (CN or AN) but large enough to capture the synoptics. For AN the latitudinal extent is set between 5° and 25°S, while for CN it is between 10° and 30°S. The longitudinal extent is the same for both regions: 5° and 22.5°E. Spatial patterns of 08:00 UTC (representative of maximum FLC dissipation; see Sect. 2.3) for ERA5 fields and half-day time averages for CAMS fields are used. Model estimation was performed using the scikit-learn package in Python (Pedregosa et al., 2011).

## 3 Results and discussion

### 3.1 Characterization of FLC and BBA during the biomass-burning season

The intensity of semi-direct effects is sensitive to the vertical layering of aerosols and clouds (Herbert et al., 2019). Figure 2 shows the CALIPSO vertical profiles of smoke and cloud layers over the Namib region. The vertical profiles were realized in two configurations. The first one is a latitudinal cross-section between 5° and 30°S within a 1° band of the coastline (Fig. 2a). The second configuration is a longitudinal cross-section between 5° and 20°E, where all latitudes between the AN and the CN (15°–24°S) are taken (Fig. 2b). The cross-sections show the well-known features of BBA and low clouds of the region (Adebiyi et al., 2015; Redemann et al., 2021). The CALIPSO observations show that typically the smoke layers are above the low-cloud layers. Most frequently, smoke layers are detected over the tropical African continent where BBA is emitted, and then transported over the South Atlantic ocean between 550 and 750 hPa. In both regions, the aerosol plumes are located higher during high BBA days, around 550 hPa, whereas on low BBA days, they are situated around 750 hPa. This pattern is likely due, for the most part, to the large-scale atmospheric processes responsible for the transport of the aerosol plumes into these regions. Both cross-sections suggest that cloud and smoke layers can be intermingling even though CALIPSO tends to overestimate the distance between the layers (Rajapakshe et al., 2017). This is particularly the case in the AN, where the BBA loading is higher and the low clouds tend to be higher up. In this situation, potential indirect aerosol effects on FLC microphysics (Costantino and Bréon, 2013; Che et al., 2022; Gupta et al., 2022) may be more relevant in the AN, even though indirect effects have also been shown further south when BBA are mixed into the marine boundary layer (Diamond et al., 2018). At the same time, mixing of BBA into the low-cloud layer can also lead to the dissipation of clouds when the absorption of solar radiation leads to a local heating (Hansen et al., 1997).

As outlined in the introduction, the strongest potential semi-direct BBA effects may be expected during the dissipation of FLCs. Figure 3 shows the observed dissipation times in the AN (Fig. 3a) and the CN (Fig. 3b) during the biomass burning season over the 2004–2018 period. To have one dissipation time per day and region, the median dissipation time is computed over all pixels for each region. Then, the data is separated into two groups of high and low BBA loading (see section 2.2) with around 200 days in each group after discarding days without FLC events, which had no data on FLC dissipation time. In agreement with Andersen and Cermak (2018), FLC dissipation is observed to mainly occur between 6 and 9 UTC. However, there is a significant difference ($p < .01$) between the FLC dissipation time on high and low BBA days, with dissipation occurring later on high BBA days, in line with the guiding hypothesis (median dissipation time 30/75 minutes later in the

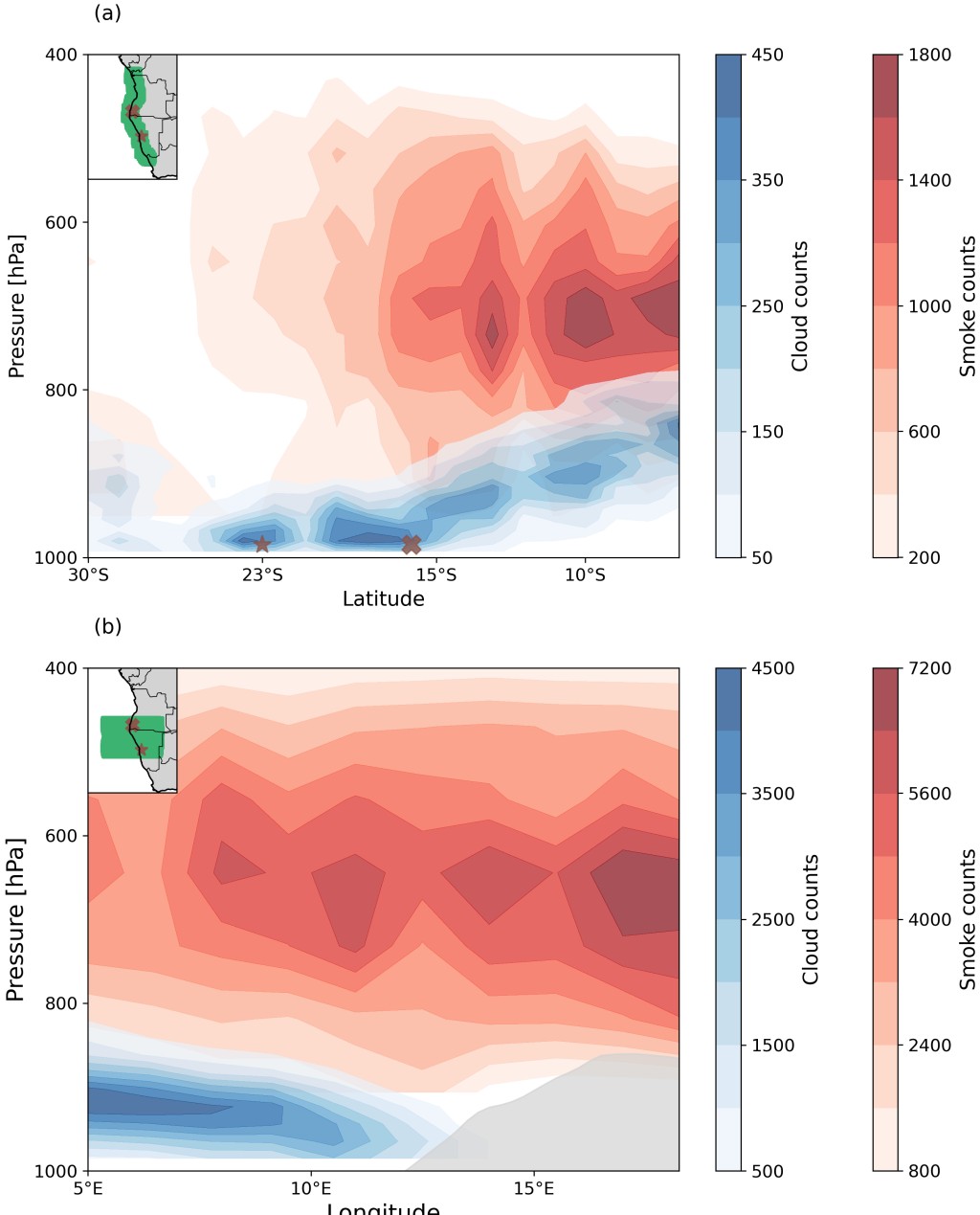

**Figure 2.** BBA season mean climatology (2004–2018) for **(a)** latitudinal cross-section of CALIOP smoke and cloud counts with longitude averages within 1° after the coastline. The brown cross (resp. star) represents the center of the AN (resp. CN). **(b)** Longitudinal cross-section of CALIOP smoke and cloud counts with latitude averages between 15°–24°S. The grey area is the mean orography within the latitude range. Only clouds below 2 km are considered. The green areas represent the ground tracks selected in each cross-section.

AN/CN). While the variability of dissipation times, measured by the interquartile range, does not change with BBA in the AN, it is 27% smaller during high BBA days in the CN, where dissipation before 7 am becomes rare. The lower number of days in the Low BBA group of the CN certainly impacts the increased variability and earlier dissipation time compared to the other groups. Additionally, the dissipation of FLCs occurs generally later in the AN than in the CN, regardless of the BBA loading.

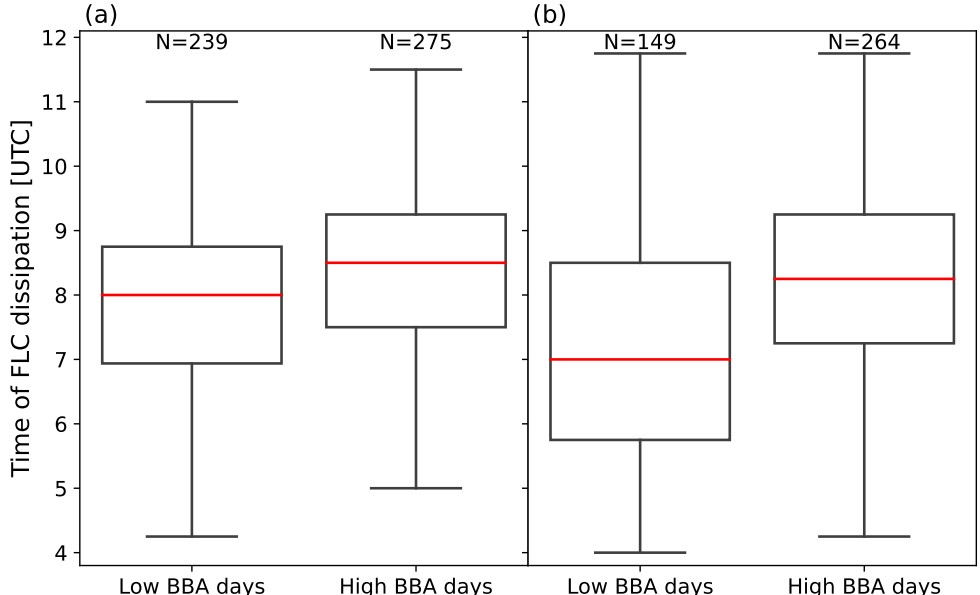

**Figure 3.** Observed time of dissipation of FLC events, for the high and low BBA days, in the AN **(a)** and the CN **(b)** during the biomass burning season in the 2004–2018 period. The red lines represent the median dissipation times, and the whiskers extend to the highest (lowest) data points still within 1.5 times the interquartile range from the third (first) quartile. FLC dissipation is significantly later ($p < .01$) on high BBA days in both regions. N indicates the number of days per group.

One should note that it is possible that separating high BBA days from low BBA days in the AN and CN may lead to sampling different meteorological situations as well because BBA occurrence over the Namibian coastline is associated with a specific recirculating pattern (Adebiyi and Zuidema, 2016). Such meteorological sampling biases would complicate the attribution of BBA effects and are thus analyzed in the following.

**3.2 Possible meteorological sampling biases**

Figure 4 shows large-scale patterns of the mean differences in Z650 and winds at 650 hPa (a, b), the typical altitude of the BBA plumes, Q650 (c, d), and BCAOD (e, f) between high and low BBA days in the AN (a, c, e) and in the CN (b, d, f). Additionally, Fig. 5 shows these differences for MSLP at 975 hPA and 10m winds (a, b), Q975 (c, d) and T2M (e, f), and EIS (g, h) in the AN (a, c, e) and the CN (b, d, f). Both figures are based on 15 years of BBA season ERA5 and CAMS (BCAOD) data. It is important to understand that the differences between the composites are temporal and not geographical: the selection of high

and low BBA days is region-specific. Z650 and winds at 650 hPa (Fig. 4a and b) show systematic differences between high and low BBA days. The continental anticyclone is stronger on high BBA days in the AN (a), leading to a clear northwesterly circulation anomaly. In the CN (b), this circulation pattern is even more pronounced, where the continental anticyclone is stronger, whereas the South Atlantic anticyclone is weaker. This circulation pattern is responsible of the large-scale advection

of moist air masses leading to marked positive Q650 anomalies on high BBA days in both regions (Fig. 4c and d). As the circulation is stronger during the CN (d) high BBA days, the Q650 anomalies extend further southward than during high BBA days in the AN (c). The BBA are also transported to the AN/CN by the same circulation, leading to similar anomaly patterns. This is coherent with the literature as BBA plumes are known to be transported in moist air masses (Adebiyi et al., 2015). The local BBA anomalies (Fig. 4e and f) on high BBA days are stronger in the AN (e), possibly because the AN is closer

to the emission sources (Swap et al., 1996), but extend further southward on high BBA days in the CN (f). The geographic locations of AN/CN may also explain the difference in the magnitude of the circulation anomaly: the proximity of the AN to the emission sources means a smaller shift in the circulation is necessary for the BBA plumes to be transported there.

Differences in MSLP (Fig. 5a and b) show negative pressure anomalies of about 1–2 hPa in the AN (a) and 4–6 hPa in the CN (b) over land during high BBA days. The near-surface wind speed anomalies show that these pressure anomalies lead to

a change in the regional coastal circulation in both regions and overall to an onshore flow anomaly. MSLP differences are of opposite sign than the Z650 differences, indicating that an anticyclonic anomaly at 650 hPa is associated with a strengthened heat low at the surface. The positive Q975 anomalies on high BBA days along the coastline in both regions are also a signal of an onshore flow of moist air masses, or rather lack of dry offshore flow (Fig. 5c and d). Figures 4 showed an advection of moist air masses in the free troposphere. These moist air masses can cause a longwave heating at the surface which can explain

the observed negative pressure anomalies (Alamirew et al., 2018). Indeed, positive T2M (Fig. 5e and f) anomalies of about 4–5 K in the AN (e) and 7–8 K in the CN (f) are observed, and their spatial patterns are clearly related to the free-tropospheric moisture anomalies. As such, the observed negative pressure anomalies are a clear sign of a heat low anomaly, which is defined as an area of low atmospheric pressure caused by intense surface heating. This phenomenon seems to be mainly driven by the greenhouse warming of the moist free-tropospheric air masses. Northerly advection of warm air masses is also observed

and likely contributes to the development of the heat low anomaly as well. It should be noted that these processes have been associated to fog occurrence in the Namib before (Andersen et al., 2020). Differences in EIS (Fig. 5g and h) show strong positive anomalies up to 6 K near the coastline in both regions. The stronger inversion during high BBA days could be related to BBA absorption and is in line with the guiding hypothesis.

The results so far indicate that at least part of the observed later FLC dissipation on high BBA days may be caused by the

transport of moist free-tropospheric air masses strengthening a continental heat low, which modifies the coastal circulation and boundary layer moisture along the coastline. In the following, possible effects of BBA on the vertical profiles of air temperature and heating rates are analyzed. Figure 6 shows differences in the temporal air temperature tendency between high and low BBA days, averaged for the BBA season over the 2004–2018 period for all pressure levels from the surface to 500 hPa for the AN (Fig. 6a) and the CN (Fig. 6b). This can be interpreted as the differences of the first derivative of the temperature. A positive

air temperature-development difference means that the air at a specific time and altitude is cooling less (during nighttime)

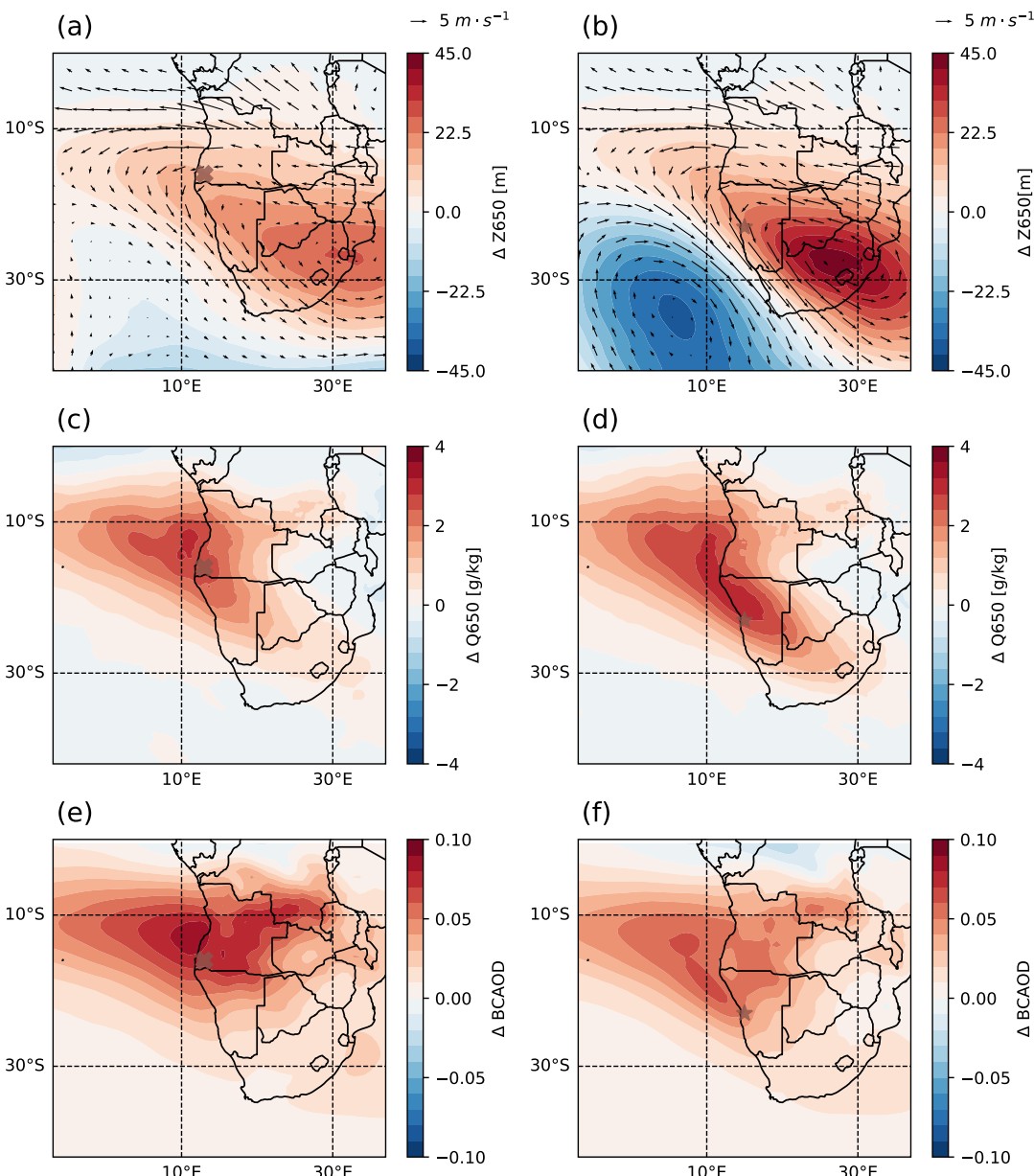

**Figure 4.** BBA season mean climatological (2004–2018) differences (high-low BBA days) for: **(a)** ERA5 geopotential height and winds at 650 hPa in the AN, **(c)** ERA5 specific humidity at 650 hPa in the AN, **(e)** CAMS black carbon AOD at 550 nm in the AN. **(b, d, f)** same as **(a, c, e)** respectively but in the CN. The **u** and **v** components of winds are bilinearly interpolated to a 2.5°x2.5° grid for clarity. ERA5 data are sampled at 08:00 UTC, while half-day averages from 00:00 to 12:00 UTC are used for CAMS. The brown cross (resp. star) represents the center of the AN (resp. CN).

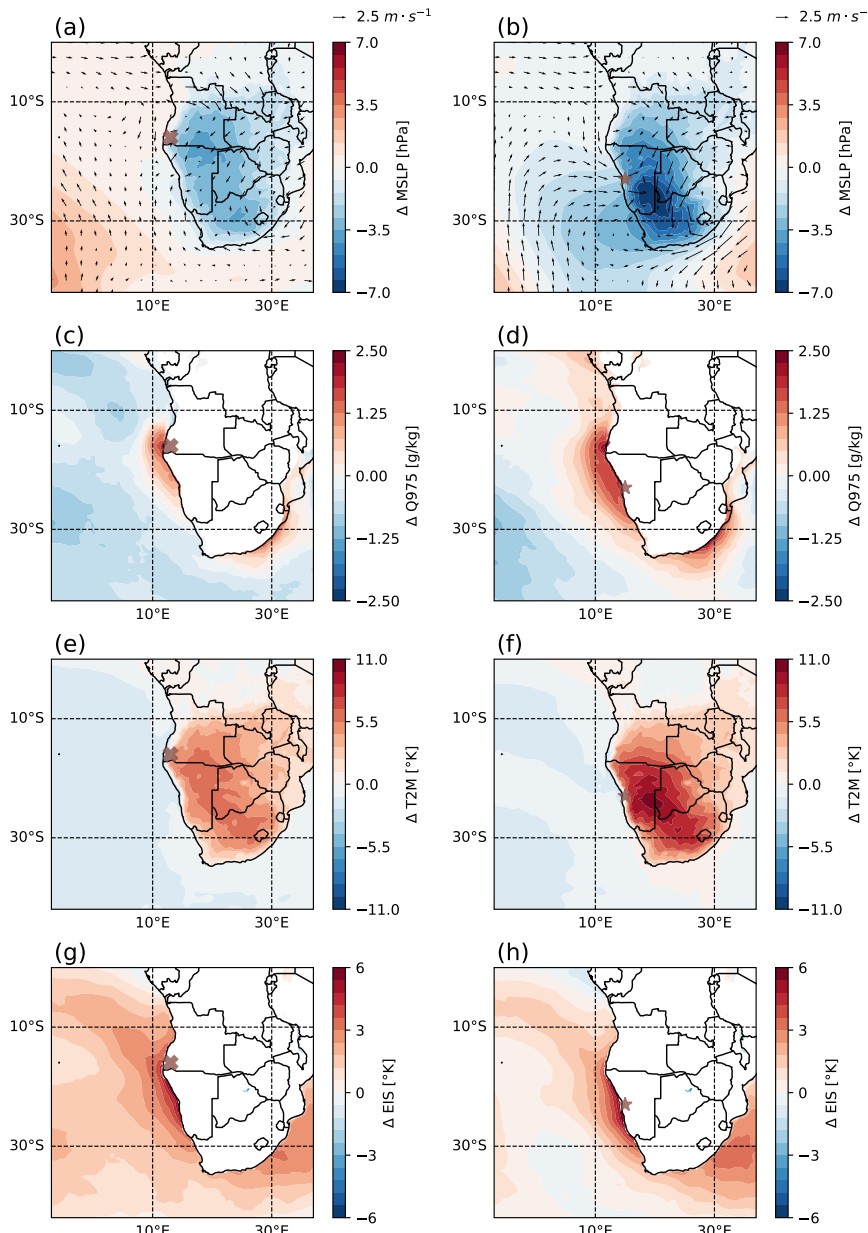

**Figure 5.** BBA season mean climatological (2004–2018) differences (high-low BBA days) for: **(a)** ERA5 mean sea level pressure at 975 hPa and 10 meter winds in the AN, **(c)** ERA5 specific humidity at 975 hPa in the AN with subsurface regions masked, **(e)** ERA5 2m air temperature in the AN, **(g)** ERA5 estimated inversion strength in the AN. **(b, d, f, h)** same as **(a, c, e, g)** respectively but in the CN. The **u** and **v** components of winds are bilinearly interpolated to a 2.5°×2.5° grid for clarity. Data are sampled at 08:00 UTC. The brown cross (resp. star) represents the center of the AN (resp. CN).

or warming more (during daytime) on high BBA days. A negative difference means that the air temperature in the grid cell is cooling less/warming more on low BBA days. In both regions, high BBA days between 4 and 8 UTC feature a weaker cooling in the boundary layer but a stronger cooling in the free troposphere. Conversely, shortly after sunrise around 8 UTC, the free troposphere features a stronger warming on high BBA days, whereas the boundary layer warming is less pronounced, particularly in the CN. A sign change of aerosol effects between night and daytime has been described before in Herbert et al. (2019). These observed differences in the vertical profiles of cooling/heating rates may be explained by the warm, moist absorbing aerosol-laden air masses in the free troposphere. During the nighttime, the warmer free-tropospheric air loses more heat and heats the surface via stronger greenhouse warming and/or reduces cloud-top radiative cooling as described in Adebiyi and Zuidema (2018). After sunrise, the absorbing BBA lead to a stronger heating in the free troposphere but may also slow down the surface heating by blocking incoming solar radiation. These differences in the temporal development of the vertical temperature structure are in line with our hypothesis and may contribute to the observed differences in FLC dissipation. One should note though that the FLC layer also feeds back to the surface temperatures and the boundary layer development via its radiative effects. In particular, the observed later FLC dissipation on high BBA days may contribute to the initially weaker boundary layer heating rates after sunrise. However, the later FLC dissipation can't explain the weaker daytime warming of the boundary layer on high BBA days, especially pronounced in the CN. But air temperatures are not only driven by radiation, there are advective contributions as well. The latter are likely to differ, particularly as coastal circulation is also changed, and cause this weaker warming.

The vertical profiles of the mean differences of air temperature between high and low BBA days at 08:00 UTC in the AN and CN (Fig. 6c) show that above 650 hPa, the temperatures are slightly warmer on low BBA days but the differences are small. Below 650 hPa the temperatures are higher on high BBA days with a maximum of 3 K at 800 hPa in the AN and 6 K at 850 hPa in the CN, so the inversion is stronger and lower-lying in the CN compared to the AN on high BBA days. This may be a potential reason for the 45 minute difference in the later median dissipation time between CN and AN in Figure 3

Additionally, the morning development of the boundary layer height (see appendix B) indicates that the planetary boundary layer (PBL) deepens slightly more until noon on low BBA days. In the CN, the PBL is marginally lower on high BBA days; however, the differences are minimal, as shown by the largely overlapping standard deviation areas.

There are distinct meteorological differences between high and low BBA days in the two regions, which are associated with the BBA transport and its local loading in the considered regions. However, by comparing situations averaged over hundreds of days, this study does not effectively capture out-of-the-ordinary events, such as mid-latitude intrusion events (Zhang and Zuidema, 2021), which can significantly impact BBA transport and distribution on specific days. While detailed case studies using back trajectories could analyze the variability of such events, they are beyond the scope of this study.

To quantify and attempt to disentangle the contributions of the relevant meteorological parameters and BBA to the observed differences in FLC dissipation time a statistical approach is used.

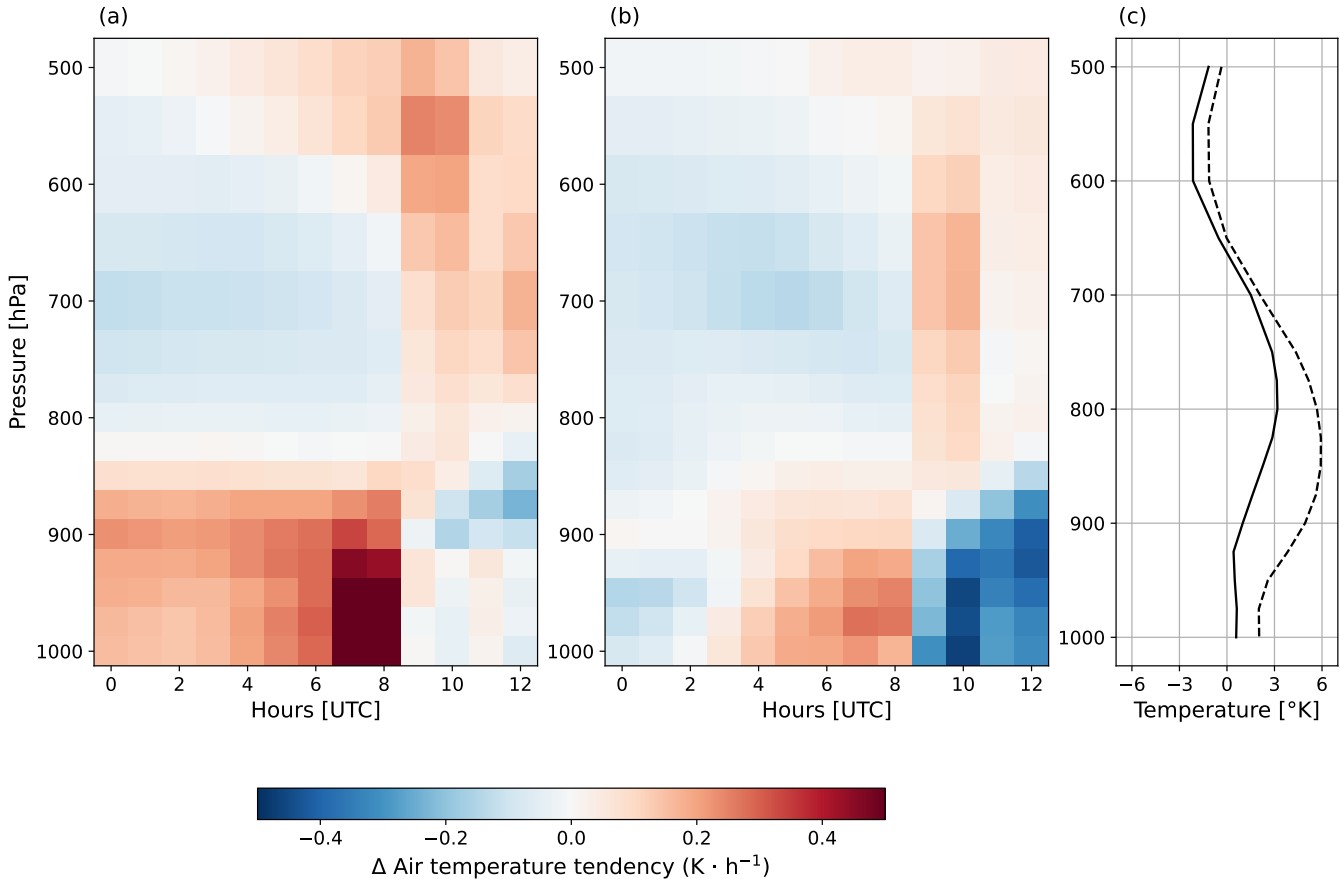

**Figure 6.** Mean differences (high-low BBA days) of the hourly tendency of air temperature ($K.h^{-1}$) at each time step for 16 pressure levels in the AN (a) and in the CN (b) during the BBA season (2004–2018). (c) Vertical profiles of the mean differences (high-low BBA days) of air temperature (K) at 08:00 UTC for 16 pressure levels in the AN (solid line) and CN (dashed line).

### 3.3 Statistical estimation of partial contributions to FLC dissipation

As described in section 2.5, spatial data of meteorology and BBA are used to predict FLC dissipation time with an ensemble of ridge regression models. The ensemble average skill in the AN is $R^2 = 0.34$ with a standard deviation of 0.04, and in the CN, it is $R^2 = 0.30$ with a standard deviation of 0.03. The plot of actual versus predicted dissipation times (see Appendix C) exhibits a relatively large spread around the line of perfect agreement, with a tendency for over-prediction in both AN and CN. Nevertheless, the highest density of points broadly follows the diagonal line. The comparatively low predictive skill likely means that important information for the prediction of FLC dissipation time is not captured by the selected predictors. For example, variability in downwelling shortwave radiation (driven by e.g. sub-seasonal variability in solar geometry, and day-to-day variability in higher-level clouds), the temporal evolution of meteorological fields and precise vertical information of BBA

would likely increase the skill of the model. Additionally, it is also possible to use individual members from the ensemble data assimilation of ERA5 as predictors to potentially increase the variability. An additional limitation of the model is the coarse resolution of the predictors can make it difficult for the model to capture local aerosol effects. Figure 7 shows ensemble mean

predicted FLC dissipation times on low/high BBA days in the AN (a) and CN (b) as boxplots similar to the observations shown in Figure 3. The variability of predicted FLC dissipation times is lower than observed, a typical sign that the statistical model is underfitting. Despite the limitations of the statistical model ensemble, the majority of the observed significant differences ($p < .01$) in FLC dissipation time between high and low BBA days can be reproduced. Dissipation of FLC is predicted to occur later during high BBA days. The median dissipation time is 35/44 minutes later in the AN/CN, so the model exactly reproduces

the observed difference in the AN but underestimates the observed differences in the CN. In the following, the mean coefficient fields of the model ensemble are used to estimate the contributions of the meteorological and BBA predictors to the predicted differences in FLC dissipation time and therefore quantify the influences of the processes outlined in section 3.2.

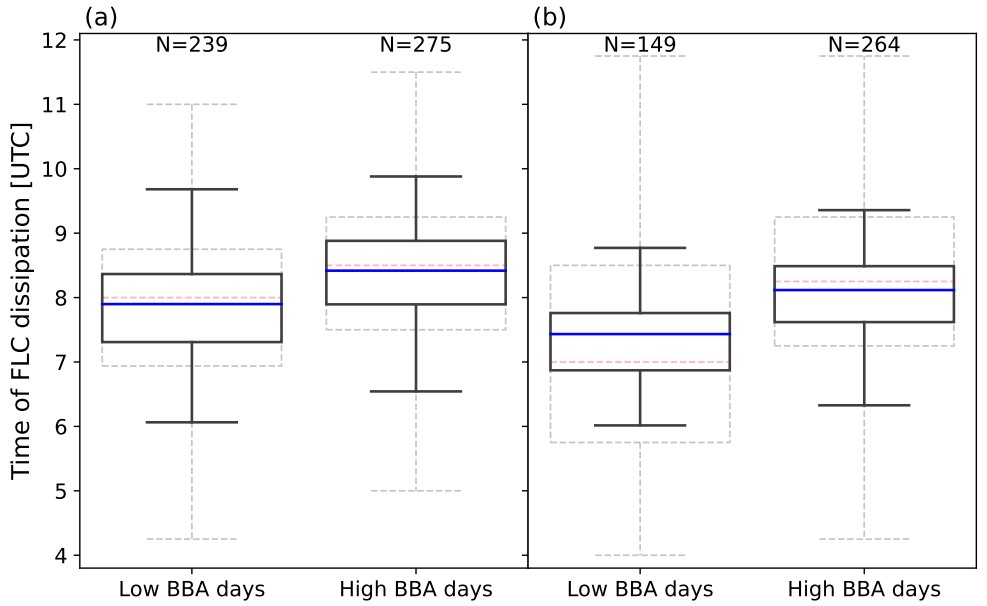

**Figure 7.** Ensemble mean predicted time of dissipation of FLC events in AN **(a)** and CN **(b)** during the biomass burning season during the 2004–2018 period. The blue lines represent the median dissipation times, and the whiskers extend to the highest (lowest) data points still within 1.5 times the interquartile range from the third (first) quartile. FLC dissipation is significantly later ($p < .01$) on high BBA days in both regions. Observed dissipation times are outlined with dashed lines in the background. N indicates the number of days per group.

Figure 8 shows the mean spatial coefficient fields for each predictor used in the model ensemble: Q975 (a, b), Q650 (c, d), MSLP (e, f), T2M (g, h), BCAOD (i, j), EIS (k, l), SST (m, n) in the AN and the CN. A clear coefficient pattern is apparent

in the regional coastal Q975 (Fig. 8a and b) in both regions. FLC dissipation time is shown to be sensitive to boundary layer moisture exactly where the anomalies on high BBA days are observed (Fig. 5). Coefficient fields of MSLP (Fig. 8e and f),

show that the coastal circulation (onshore advection of moist air masses in the boundary layer) is not only important for the occurrence of FLCs (Andersen et al., 2020), but also for its lifetime. Near-surface air temperatures (Fig. 8g and h) also show a clear spatial coefficient pattern, with negative coefficients along the coastline and positive coefficients further inland. These patterns may describe the radiative feedback of FLCs on near-surface air temperatures along the coast and the contribution to the heat low further inland. For BCAOD (Fig. 8i and j) the coefficient pattern is fairly noisy and does not give a clear signal. Nevertheless, there is a very localized positive sensitivity of FLC dissipation time to EIS (Fig. 8k and i) which in turn may be partially driven by BBA absorption (Fig. 6). The coefficient pattern of SST (Fig. 8o and p) suggests that colder temperatures in isolated coastal regions may be related to delayed FLC dissipation times.

By multiplying the ensemble mean coefficients shown in Fig. 8 by the mean predictor fields on high and low BBA days, the spatial contributions of each predictor to the predicted FLC dissipation times (Fig. 7) are obtained. Subtracting the contributions on low BBA days from those on high BBA days results in the contributions to later dissipation times during high BBA days. This is shown by Fig. 9 for the same predictors used in Fig. 8 in the AN and the CN.

A strong positive contributor to later dissipation times is Q975 in both regions (Fig. 9a and b), due to strong positive sensitives of the model (Fig. 8) and positive Q975 anomalies (Fig. 5). Despite positive anomalies in the PBL, the coefficient field for Q650 (Fig. 9c and d) is noisy in both regions and so are the contributions. Nevertheless, there are positive contributions over land, where a potential greenhouse warming may lead to strong warming contributing to the heat low and thus onshore circulation anomaly. Over the ocean, the negative contributions could be attributed to a reduction in the cloud-top radiative cooling by the moist free troposphere (Zheng et al., 2018), which reduces the mixing of surface moisture into the FLC (Caldwell et al., 2005). For MSLP, there is a strong positive contribution in the CN (Fig. 9f) and positive but weaker contributions in the AN (Fig. 9e). The difference in the magnitude of contributions between the AN and the CN can be explained by the stronger anomalies observed in the CN in section 3.2 and the stronger circulation needed because the CN is farther away from BBA sources than AN. T2M is related to moisture in the free troposphere via greenhouse warming. In the case of the AN model (Fig. 9g) along the coastline, the negative sensitives coupled with the negative T2M anomalies result in positive contributions. Whereas, in the case of CN (Fig. 9g), the model's stronger inland positive T2M anomalies lead to weaker coastline negative anomalies and result in negative contributions. On the other hand, in both regions, a heat low leads to positive inland T2M contributions. BCAOD contributions are strong but noisy (Fig. 9i and j). Additionally, in this study, BCAOD is the parameter used to create the high and low BBA groups, therefore the relative differences between the groups are particularly large, amplifying the noisy coefficient fields. For these reasons the robustness of this contribution is questionable. On the other hand, the very localized but strong positive contribution of EIS in the AN (Fig. 9k) could be related to absorbing BBA layers leading to local heating in the free troposphere and thus a stronger inversion. In this perspective, the weaker contributions in the CN (Fig. 9i) could be explained by the longer distance of the region from the BBA sources and smaller concentration of smoke layers as seen in Figure 2a. Finally, SST has positive contributions in the AN (Fig. 9m). These contributions are the product of a negative coefficient field and a negative SST anomaly near the coastline (not shown). This could partly be explained by reduced incoming solar radiation due to absorption in the free troposphere. However, since the ocean surface responds only slowly, other factors, such as the changed circulation possibly contributes as well. Mallet et al. (2024), a recent model-based study which finds that

an SST decrease, associated with solar heating in the free troposphere, is responsible for a positive feedback of BBA radiative effects on low-level clouds. In the CN (Fig. 9n), contributions are weaker and noisier and the SST anomaly is spatially removed from the CN and the BBA anomaly. Overall, according to the statistical model, increased moisture in the PBL and changes in coastal circulation, with an onshore anomaly, are the main causes of later FLC dissipation which may be a bit counterbalanced by the moist free troposphere air masses initiating FLC dissipation by reducing cloud-top radiative cooling over the ocean.

## 4  Conclusions and outlook

The central objective of this study was to investigate to what extent the circulation of the seasonally occurring biomass burning aerosols (BBA) may influence the dissipation time of fog and low clouds (FLC) events in the Namib region, and more precisely in two sub-regions with a high frequency of FLC events: the Central Namib and the Angolan Namib. To do this, a novel satellite-based data set of FLC dissipation times was used. In addition, ERA5 reanalysis of meteorological fields and CAMS black carbon AOD products were analyzed. For each of the two regions, two groups of days with high and low BBA loading were created to analyze possible BBA effects on FLC dissipation time. The main findings of this study are:

1. During the BBA season (June–October) of the investigated 15-year period (2004–2018), FLC dissipation time is significantly later on high BBA days in both regions. This is consistent with our guiding hypothesis and with recent studies, which have shown that BBA layers tend to increase cloudiness (Wilcox, 2010; Sakaeda et al., 2011; Gordon et al., 2018; Deaconu et al., 2019). But these studies were focused on the South Atlantic, here this knowledge is extended over land. However, the grouping of high and low BBA days has been shown to lead to meteorological sampling biases, complicating the separation of meteorological effects from the "large-scale" semi-direct BBA effects.

2. BBA are shown to be transported along moist free-tropospheric air by a large-scale circulation pattern. The associated longwave heating causes a continental heat low, which modifies the coastal circulation and boundary layer humidity along the coastline. A longwave heating modulating coastal circulation has been shown to drive fog occurrence (Adebiyi et al., 2015; Adebiyi and Zuidema, 2018; Andersen et al., 2020). Here, the understanding of the system is expanded by showing it is also important for the dissipation of FLCs. Additionally, this study shows that similar to the South Atlantic (Adebiyi et al., 2015; Adebiyi and Zuidema, 2018; Deaconu et al., 2019), longwave heating impacts FLCs over land as well.

3. The temporal development of vertical temperatures shows a stronger nighttime cooling and surface/boundary layer warming during high BBA days. After sunrise, the free troposphere features stronger warming, whereas the boundary layer does not warm as quickly during high BBA days, strengthening the local inversion. Because aerosol effects have been shown to change sign after sunrise (Herbert et al., 2019), the differences in temporal temperature evolution are thought to be driven by the greenhouse effect of moisture in the BBA plumes and BBA absorption. These factors may contribute to the observed differences in FLC dissipation.

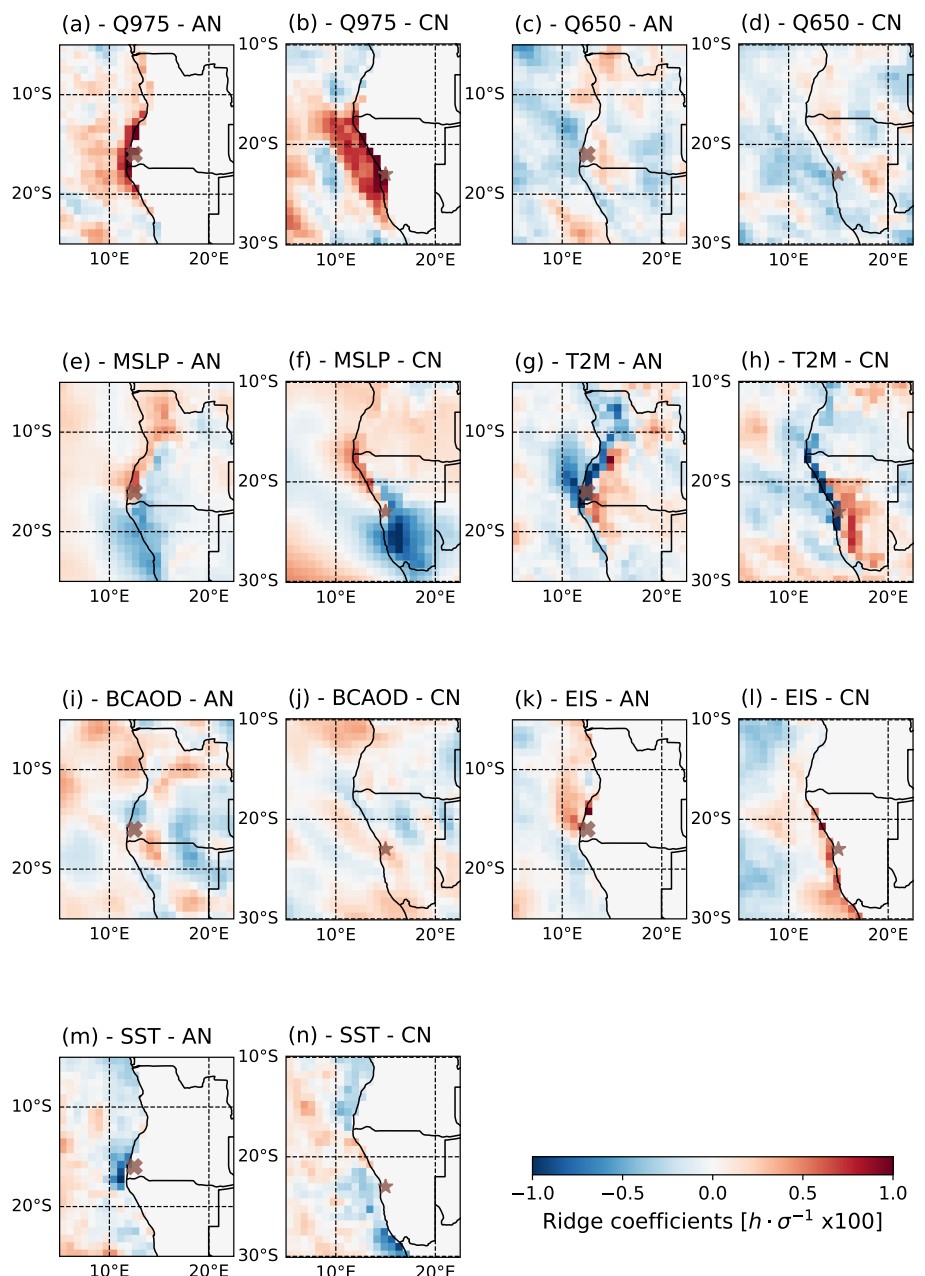

**Figure 8.** Ensemble mean spatial coefficient fields (hours per standard deviation x 100) for each predictor:**(a, b)** specific humidity at 975 hPa with subsurface regions masked, **(c, d)** specific humidity at 650 hPa, **(e, f)** mean sea level pressure, **(g, h)** 2m temperature, **(i, j)** black carbon AOD, **(k, l)** estimated inversion strength, **(m, n)** sea surface temperature. In the AN **(a, c, e, g, i, k, m, o)** and the CN **(b, d, f, h, j, l, n, p)**. The brown cross (resp. star) represents the center of the AN (resp. CN).

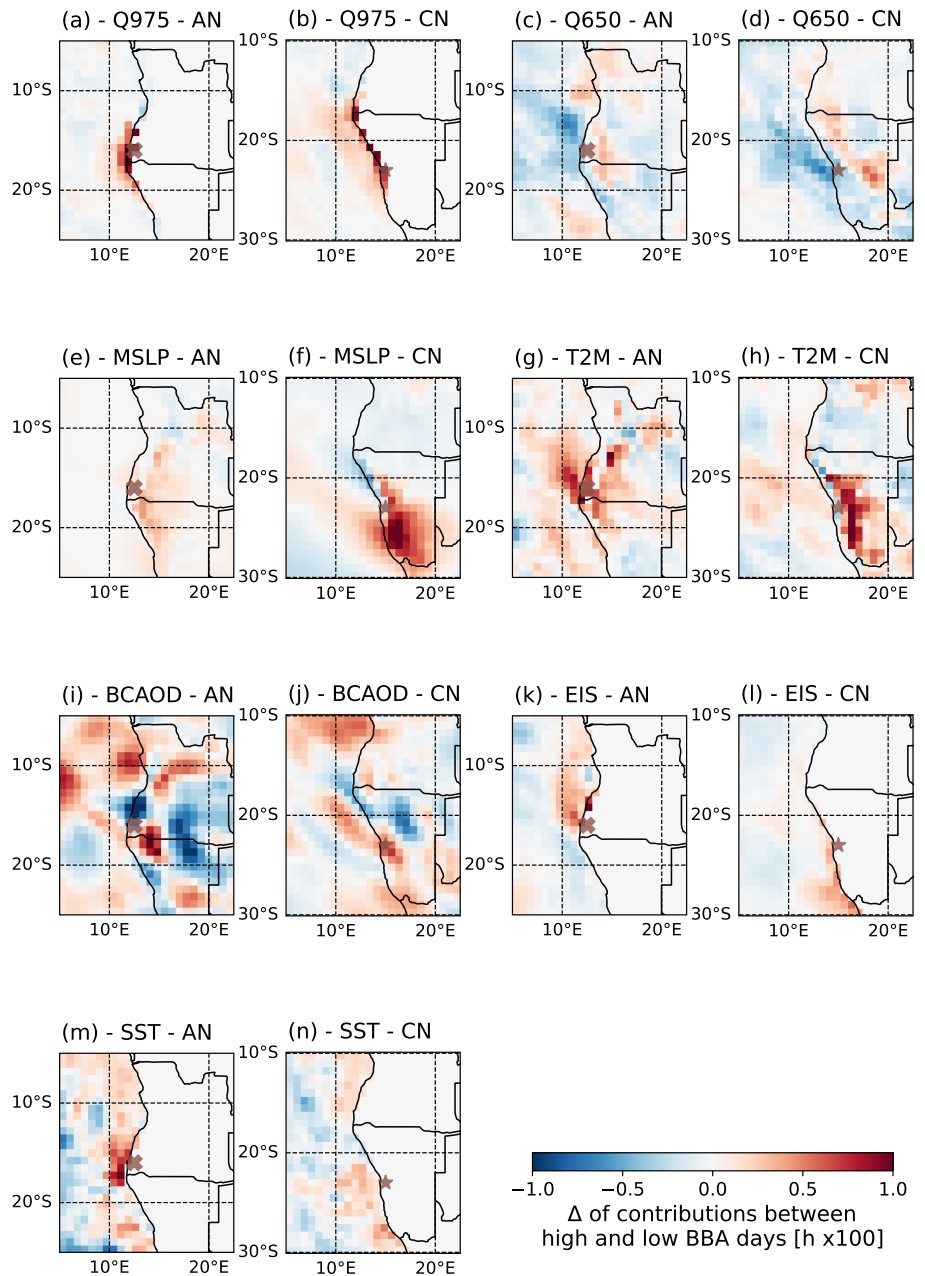

**Figure 9.** Ensemble mean differences (high-low BBA days) of spatial contributions (hours x100) for each predictor:**(a, b)** specific humidity at 975 hPa with subsurface regions masked, **(c, d)** specific humidity at 650 hPa, **(e, f)** mean sea level pressure, **(g, h)** 2m temperature, **(i, j)** black carbon AOD, **(k, l)** estimated inversion strength, **(m, n)** sea surface temperature. In the AN **(a, c, e, g, i, k, m, o)** and the CN **(b, d, f, h, j, l, n, p)**. The brown cross (resp. star) represents the center of the AN (resp. CN).

4. A statistical model (ridge regression) was used to quantify and partially disentangle meteorological and BBA effects. The statistical model is shown to be able to reproduce the observed differences in FLC dissipation time on high and low BBA days and attributes these differences mainly to differences in circulation, boundary layer humidity and near-surface temperatures along the coastline. With this model, it is not possible to have definitive conclusions about BBA effects, as the underfitting makes it hard to capture subtle semi-direct BBA effects on FLC dissipation. Additionally, using this model requires the assumption of linearity in the underlying processes. Despite these limitations, the contributions of EIS, for example, may hint at semi-direct BBA effects.

The findings highlight the difficulties of disentangling meteorological and aerosol effects on low cloud development through statistical modelling approaches. While most of the differences in FLC dissipation time are likely driven by meteorology, some observational indications of possible semi-direct effects on FLC dissipation are found. To further investigate and fully disentangle meteorological and BBA effects, targeted analyses with large-eddy simulations are essential. Additional approaches to better constrain meteorological influences can be employed, such as selecting air masses with similar hydrologic histories using isotope observations (Henze et al., 2023), computing backward trajectories of air masses with comparable dynamical and thermodynamical conditions (Andersen et al., 2020) but contrasting BBA loadings. Additionally, radiative sensitivity studies such as Obregón et al. (2018), could also be useful in disentangling aerosol direct effects from meteorological covariates. New satellite data from cutting-edge instruments, such as the High Spectral Resolution Lidar onboard the EarthCARE satellite (Wehr et al., 2023), will improve our understanding of aerosol and cloud interactions. In a different approach, one could use a statistical model, similar to the one developed here, with CMIP6 data as predictors, as done by Ceppi and Nowack (2021), to assess the impacts of climate change on FLCs in the region.

# Appendix A: Dissipation time and AOD distributions

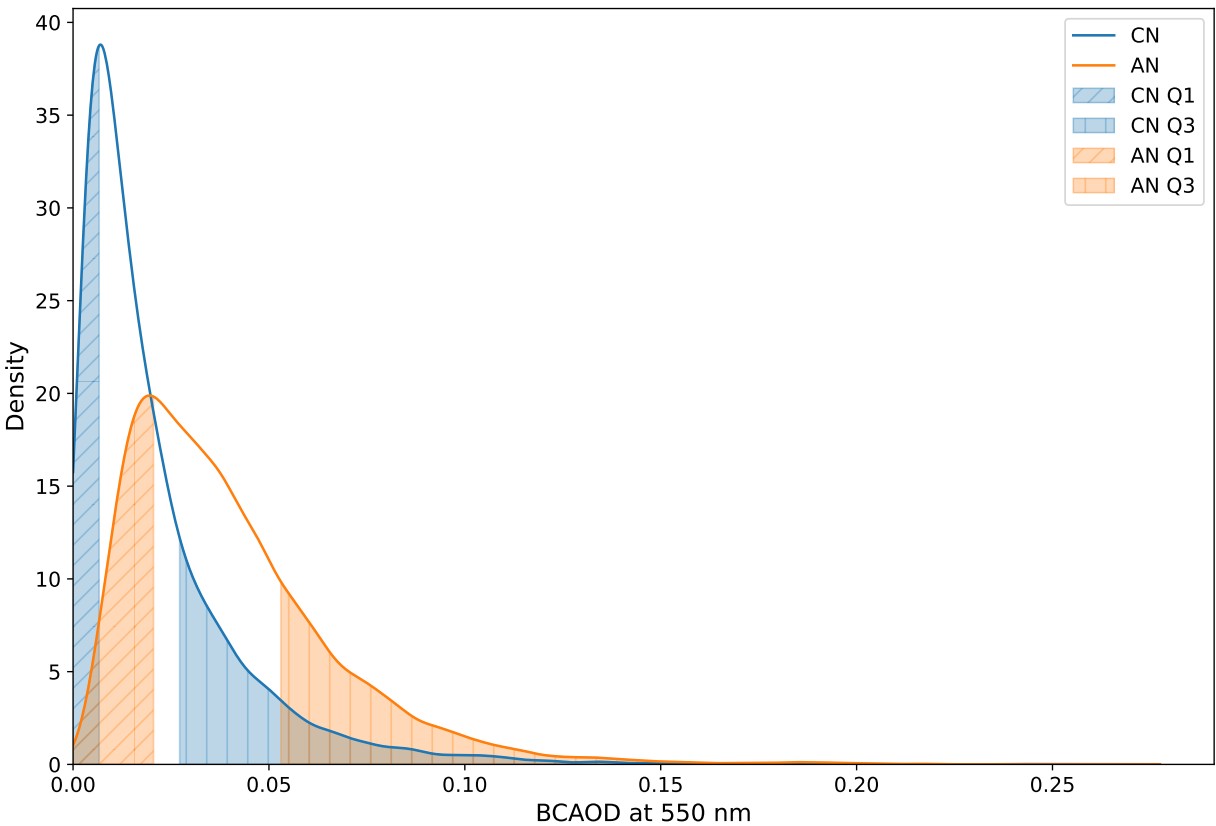

**Figure A1.** BBA season mean climatology (2004–2018) of the BCAOD probability density distribution in the AN and CN, along with their respective 1st and 3rd quartiles.

**Appendix B:  Morning boundary layer height development**

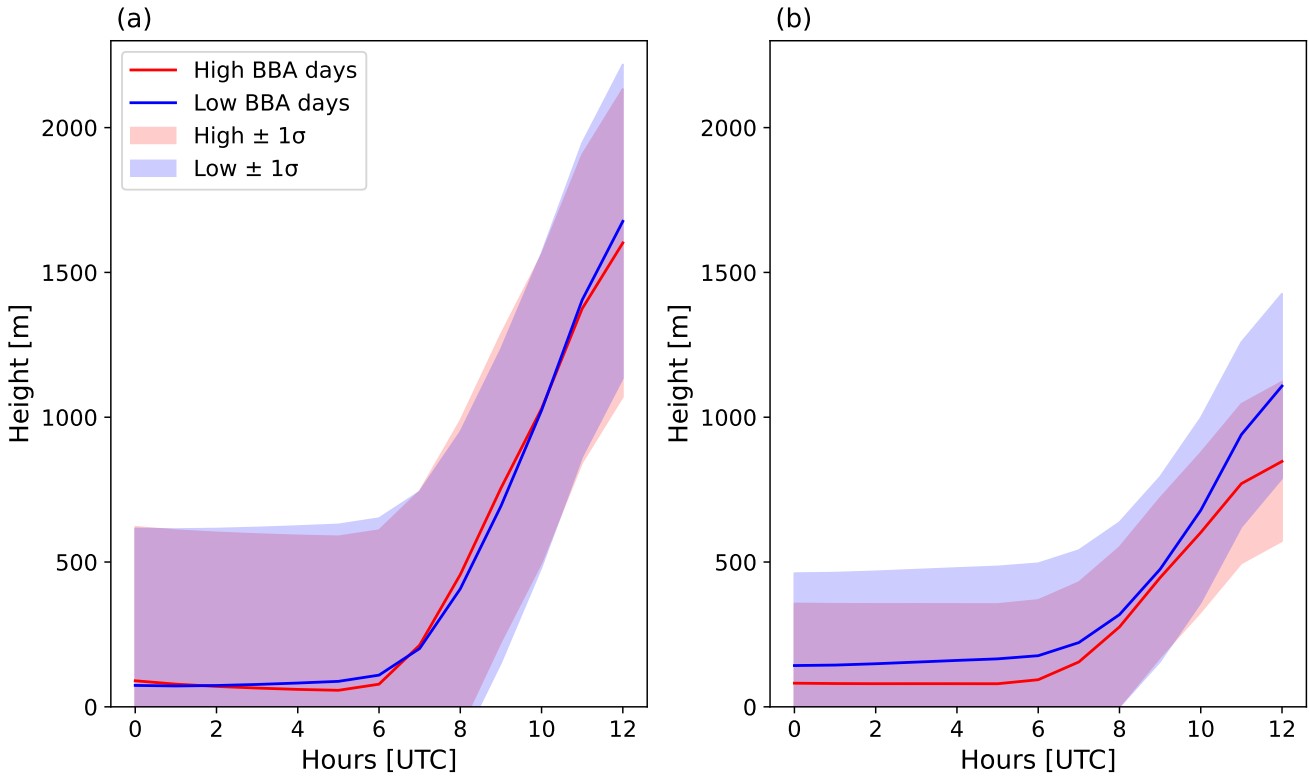

**Figure B1.** BBA season mean climatology (2004–2018) of the mean hourly boundary layer height in the AN **(a)** and the CN **(b)**.

**Appendix C: Prediction versus truth scatter plot**

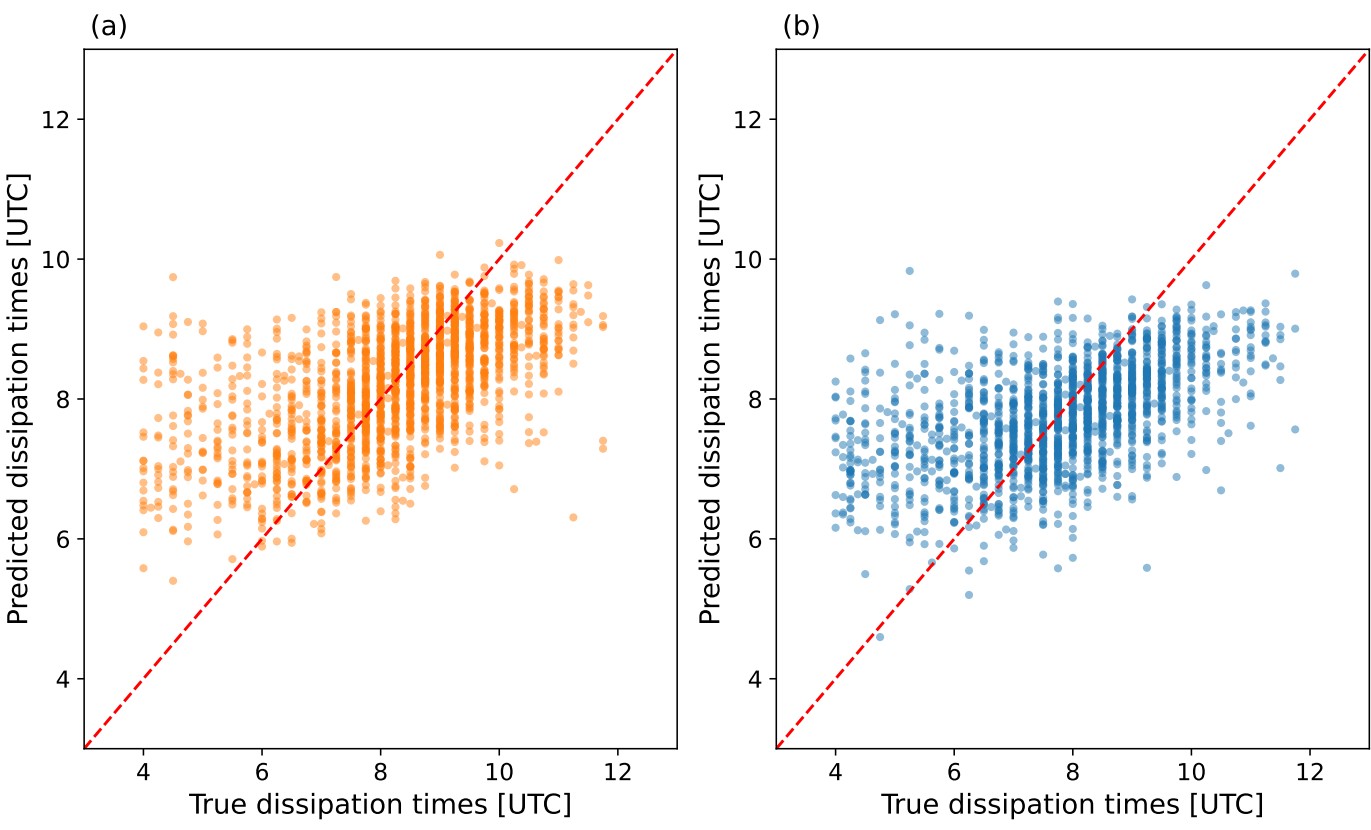

**Figure C1.** Ensemble mean predicted vs. true values for dissipation times [UTC] for the AN (a) and CN (b) models. The red dashed line represents the y = x line, indicating where predicted values equal true values.

*Code and data availability.*   CAMS data were generated using Copernicus Atmosphere Monitoring Service. ERA5 data were generated using Copernicus Climate Change Service Information. CALIPSO data were generated using the CALIPSO Search and Subsetting web application. Satellite FLC data is available with DOI: 10.35097/z5adwk39bbfke404. Code for data processing is available from the corresponding author upon reasonable request.

    *Author contributions.*   AM, HA and JC had the idea for the analysis. AM obtained and analyzed most of the data sets, conducted the original
research, and wrote the manuscript. EP contributed to the design and creation of the satellite FLC data set. HA, JC, PF, EP and JQ contributed to article preparation and the interpretation of findings.

*Competing interests.* The authors declare that they have no conflict of interest.

*Acknowledgements.* Funding for this study was provided by Deutsche Forschungsgemeinschaft (DFG) in the project Namib Fog Life Cycle Analysis - Aerosols and Climate (NaFoLi[CA]$^2$), project number 462604610. Paola Formenti was funded through the AEROCLO-sA project, supported by the French National Research Agency under grant agreement n° ANR-15-CE01-0014-01, the French national program LEFE/INSU, the Programme national de Télédetection Spatiale (grant n° PNTS-2016-14), the French National Agency for Space Studies (CNES), and the South African National Research Foundation (NRF) under grant UID 105958, the European Union's 7th Framework Programme (FP7/2014-2018) under EUFAR2 contract n°312609. The contribution of Julian Quinting was funded by the European Union (ERC, ASPIRE, 101077260). We thank Martina Klose for discussion and ideas which improved the study. We are also grateful to the editor, Franziska Aemisegger, and the two anonymous reviewers for their careful and constructive feedback, which has helped improve the manuscript.

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
