# Peer review of "A satellite-based analysis of semi-direct effects of biomass burning aerosols on fog and low cloud dissipation in the Namib Desert"

_EGUsphere, 2024_

## Referee Comment (RC1)

**Review of "A satellite-based analysis of semi-direct effects of biomass burning aerosols on fog and low cloud dissipation in the Namib Desert" by Alexandre Mass et al.**

This manuscript assesses the impact of seasonally emitted biomass burning aerosol (BBA) on the fog and low cloud (FLC) dissipation in the Namib Desert, using long-term geostationary satellite observations, reanalysis data, and a statistical model tasked to disentangle the role of BBA from that of the co-varying synoptic-scale meteorological pattern. They find that the FLC dissipation time is significantly later on high BBA days, which is mainly attributed to the longwave radiative effect of the co-transported moisture and subsequent changes in the regional circulation pattern and atmospheric heating profiles. Although the ridge regression model they trained is able to reproduce the statistical mean difference in the FLC dissipation time between high and low BBA days, it fails to provide definitive conclusions about BBA effects, due to the underfitting issue.

The manuscript is well written and enjoyable to read. I find this work appears of sound methodology and is of great interest to the community, with particular implications for the climate and the hydrological cycle of the Namib Desert.

I do have a few minor points/comments on the paper that I would like the authors to consider and address, in order to improve the manuscript's clarity and the soundness of the conclusions.

**Comments:**

- Attribution to BBA semi-direct effect. To me, the evidence that the authors demonstrated for a dominant role of the changing meteorological patterns (circulation, heating profile, moisture LW effect) in delaying FLC dissipation time is convincing and robust, but there are two subtleties to this that I think the authors could address/discuss in the paper
    o The definition of 'semi-direct' effect of BBA, one thing I think this paper wasn't very clear about is the definition of the 'semi-direct' effect (appears in the title) or BBA effect, one could argue it's the local effect of BBA on FLC, *all else equal*, or, one could define it as the net, integrated impact of the presence of BBA on FLC (accounting for large-scale circulation adjustments). For example, Diamond et al. (2022) discusses both the "large-scale" and "local" semi-direct effects of BBA on low clouds in the SE Atlantic.
    o Changes in meteorological conditions and coastal circulation that the authors focus on discussing in the paper, to me, is part of this "large-scale" BBA/moisture semi-direct effect. However, there could be potential contributions simply from the spatiotemporally correlated synoptic patterns and regional BBA transport, which has nothing to do with the moisture LW effect or the BBA absorption, one such example is the mid-latitude intrusion events (more frequent in Sept.) that can constrain the smoke plume closer to the continent (e.g., Zhang & Zuidema, 2021, ACP). I wonder if the method used in this study can indicate/control such correlations?

- Clarity on the methodology.
    o It wasn't clear to me whether only June-October are used in the analyses or the whole year was used? (L82 says 15 years of data are used while L344 says only months June-October are used)

- L83 'some of the analyses' and L128 'For some analyses,' are not clear, please clarify and be specific.
- L103, please briefly summarize how does this work, such that one doesn't need to go to Pauli et al. (2022) to grasp the idea of this method (one can of course read it if more details are sought).
- What are the predictors used in the ridge regression? It says broadly "the predictors are the spatial fields of ERA5 meteorology and BCAOD from CAMS" I assume all the variables mentioned in section 2.3 are included in the training?
- What exactly are the thresholds for BCAOD ($25^{th}$ and $75^{th}$ percentile)? (a histogram as a supplementary figure would be nice)
  - How good is this column-integrated BCAOD reanalysis, since there are aircraft measurements from the field campaigns in the region, I wonder if this product can be validated against observations?
  - I wonder if the authors have considered using the above-cloud AOD (ACAOD, by Kerry Meyer at NASA) product to indicate BBA loading, given it's more observationally-based.
  - L120, assuming 15 years of Jun-Oct days are used, a $25^{th}$ percentile will yield ~562 days for each 'high BBA' and 'low BBA' group, is there additional screening involved? or missing data?
  - L194, why only 200 days? instead of 300 days (L120)
- In section 2.4, CALIPSO is introduced, but I don't see how it is used. I couldn't find anywhere in Section 2 or 3 where CALIPSO data is used/discussed. L279 states the lack of skill of the statistical model could be due to the lack of vertical information of BBA (it confuses me as I thought you used CALIPSO to get the vertical dimension).

- Statistical model
  - In general, I wonder what's the rationale to stick to this ridge regression *linear* model (given the low $R^2$ values), over other *non-linear* models, such as CNN or random forest?
  - What's the ensemble spread in terms of member skill and the prediction of the mean delay in dissipation time?
  - Could you show the scatter plot between truths and predictions for this model?
  - As mentioned in L279, have the authors tried other predictors to try to improve the skill of the model?

**Editorial:**

- L103, what is 'logistic regression'? a typo?
- L133, do we know which direction?
- Figure 2, cross and star labels on the map is reversed?
- Figure 3, caption, please define IQR at first use.
- L230-235, what's the reason for such a strong land-sea contrast (reversed in the sign) in terms of T2M difference between high and low BBA days?
- L349, as mentioned in my first comment, BBA effects can include these large-scale circulation adjustments, depending how one defines it.

- L366, perhaps, given the primary role of BBA - modulating the large-scale synoptic pattern/circulation, the use of this regression model is not suited?
- L375, one possible way of doing this could be the "synoptic matching (or locking)" method used in Quaas et al. (2021, ERL), where, given a location and time, they search in their climatological database for a day with the synoptic pattern that matches the current one the best, but with different aerosol states.
- L375-377, given that CALIPSO is used (?) already in this study and that the key issue is the covariation between meteorology and BBA, I struggle to see how can EarthCARE offer better ways to disentangle aerosol effects.

**References**

Diamond, M. S., Saide, P. E., Zuidema, P., Ackerman, A. S., Doherty, S. J., Fridlind, A. M., Gordon, H., Howes, C., Kazil, J., Yamaguchi, T., Zhang, J., Feingold, G., & Wood, R.: Cloud adjustments from large-scale smoke–circulation interactions strongly modulate the southeastern Atlantic stratocumulus-to-cumulus transition. Atmospheric Chemistry and Physics, 22(18), 12113– 12151. https://doi.org/10.5194/ACP-22-12113-2022, 2022.

Zhang, J. and Zuidema, P.: Sunlight-absorbing aerosol amplifies the seasonal cycle in low-cloud fraction over the southeast Atlantic, Atmos. Chem. Phys., 21, 11179–11199, https://doi.org/10.5194/acp-21-11179-2021, 2021.

Quaas, J., Gryspeerdt, E., Vautard, R., and Boucher, O.: Climate impact of aircraft-induced cirrus assessed from satellite observations before and during COVID-19, Environmental Research Letters, 16, 064051, https://doi.org/10.1088/1748-530 9326/ABF686, 2021.

---

## Referee Comment (RC2)

GENERAL COMMENTS

Fog is an important source of moisture in arid regions, and thus, the question of when and where fog is present is of high importance. This well-written paper investigates how biomass-burning aerosols (BBA) influence the (diurnal) dissipation time of fog/low cloud (FLC) events in Namib. Various satellite data and reanalysis data serve to analyze differences between high and low biomass burning conditions revealing significant differences also in respect to the meteorological conditions. Finally, a statistical model is built to predict the dissipation time from meteorological fields, showing the difficulty of disentangling aerosol and meteorological effects. The paper is certainly interesting for ACP and of high quality, however, I have several comments/ideas for further improvement.

- The area of investigation is rather void of observations. Therefore, reanalysis highly depends on satellite data, which have problems in resolving the boundary layer, and the underlying NWP model. Because the 4D-Var data assimilation uses 12-hour windows (from 09 UTC to 21 UTC), jumps can occur at these times in parameters (such as water vapor) that are not constrained strongly by observations. This might influence the results in Fig. 6 as the largest changes exactly occur around the 9 UTC time. As an example, you can find the diurnal cycle of total column water vapor for Iquique, Chile, at the coast of the Atacama. Here the 21 UTC jump is most pronounced. I find the physical explanation for the behavior sound, but it would be good to check the consistency of your data and mention ERA5 limitations in the manuscript.

- I like the classification into low and high BBA conditions, which have, by definition of the percentiles, the same number of members. However, I am wondering whether one could be even more successful with an event-based approach, as I suspect that the BBA effect is most effective during selected episodes when filaments of moist and polluted air arrive in the target regions. Especially for the southern CN region satellite total column water vapor images sometimes hint at this. Therefore, it would be good to see the frequency distribution of CAMS BCAOD to illustrate how the 25 and 75 percentiles are derived and also how they differ for both regions.

- Coming back to my suspicion that the strong pollution mainly occurs as episodes. This counteracts the assumption of a linear model to predict FLC times. Why didn't the authors perform trajectory calculations (such as in Andersen et al. (2020) to check the meteorological similarity of the different BBA events?

- The target of the study is the dissipation time. Therefore, it would be good to provide more information on its variability, not only the aggregated statistics (boxplot) from Fig.3, especially as this variability is not well reproduced by the regression model (Fig.7). For these figures and all further analysis two geographical regions (AN, CN) are considered over which averages are provided. Even after looking at the study by Anderson and Cermak (2018) the motivation to put everything together in the two regions was not clear for me. Why not look at continuous development as a function of latitude and similarly as a function of distance from the coast (which the authors themselves mention to have a strong influence)? Maybe this information could be shown graphically to motivate the choice of the two regions. In this respect, it is also not clear to me how clear sky days are treated, e.g. coastal clearings.

- The last paragraph of the introduction contains the hypothesis states that BBA events modify the inversion and the early morning development of the planetary boundary layer. Partly, this is nicely shown in Fig. 6) but what about other parameters, such as boundary layer height or water vapor (as Q975 is so important in the ridge regression)? Vicencio et al. (2023, Fig. 10) showed that the Namib has an especially high variability in boundary layer height in austral summer. Therefore Q975 might sometimes be in the BL and sometimes not.

SPECIFIC COMMENTS

- Abstract. Mention that you are looking at the time of day when the fog is dissolving. The reader could also think that dissipation time is the duration of a fog event.

- Introdcuction: please briefly explain the diurnal coastal circulation

- L25 better represent instead of resolve

- L74 define semi-direct

- L74-79: Here, it would be good to mention briefly how you want "to disentangle the BBA effects from other meteorological covariates."

- L97: Does 97% mean the hit rate excluding false positives? Or the CSI?

- Section 2: I had difficulties to extract information on data amount and resolution.
  Amount of data: June to October is roughly 150 days; with 15 years, this makes 2250 days. However, you have about 200 samples in one quartile, making the total sample about 800 days. What about the rest?
  Resolution of SEVIRI: My assumption is that each area has a grid (about 6 pixels in longitudinal and many in latitudinal direction). For each gridpoint, you have (at most) one dissipation time per day. Is this correct? If yes, then it may help to delete the 15-minute resolution in line 109.  It would also help to say in Fig. 3 how many data points are going into each bar. have: are available

- Fig.2 Why don't you show this separately for the high and low BBA situations? This would strongly help to better understand the differences between the ABN and CN region.

- L117: Please motivate why early morning.  – this only comes later in L128. At this time the reader does not know that early morning is the most frequent dissipation time

- L139: Please explicitly say in contrast to all other data another time range is used due to…-

- L170: Is the area large enough to encompass the important transport aspects? To make it easier for the reader, It would be good to mark the two regions in Fig. 4

- L163: What is actually the resolution of your predictand. Is it the average FLC in one region per day or do you do it for each satellite pixel? I assume the average as you compare it in Fig.7 with the observed medium dissipation time? Still, this could have effects on the variability of FLC, which is poorly predicted.

- L164: It might be lengthy, but it is good to have the list of variables of the "spatial fields of ERA5 meteorology" used as predictors. Is subsidence included?

- L198: Mention the definition of variability,  IQR?

- L206: Motivate 650hPa. It is done later but before wrote that the transport is highest in 650 hPa (L183). Z and Q are not defined,

- L247: grid cell is better than pixel here

- L231: T2M not defined

- L275: A correlation coefficient of 0.58 (explained variance 0.34) is not too bad. Did you try to identify the cases when especially poor predictions were made, e.g. by colouring scatter diagrams etc?

- L323:"These contributions are due to a negative SST anomaly close to the coastline which may be explained by the greenhouse warming in the free troposphere". Not clear to me..

- Conclusions: Wouldn't radiative sensitivity studies also be helpful in disentangling water vapor and BBA contributions?

TECHNICAL CORRECTIONS
- L 160: Greek letter for lambda
- I find that the readability of the text is degrading when figure caption information (that is not important for following the text) is repeated in the test. My recommendation would be to remove -statements such as in L210 (Additionally, in the Q975 fields (Fig. 5c and d) the subsurface regions at this pressure level are masked).  Or statements such as like "are shown in the third panel.", "dashed lines".
- L330. Blank missing

**Reference**

Vicencio, J., C. Böhm, J.H. Schween, U. Löhnert, and S. Crewell, 2023: A comparative study of the atmospheric water vapor in the Atacama and Namib Desert, Global and Planetary Change, 104320, https://doi.org/10.1016/j.gloplacha.2023.104320.

[Figure]

**TCWV (hourly average minus daily mean) from 1994-2023**
**Data from ERA5 at Iquique [Atacama]**

---

## Author Comment (AC1)

**Response to comments from reviewer 2**

**Manuscript:** A satellite-based analysis of semi-direct effects of biomass burning aerosols on fog and low cloud dissipation in the Namib Desert                                                              # EGUSPHERE-2024-1627

**General comments**

*This manuscript assesses the impact of seasonally emitted biomass burning aerosol (BBA) on the fog and low cloud (FLC) dissipation in the Namib Desert, using long-term geostationary satellite observations, reanalysis data, and a statistical model tasked to disentangle the role of BBA from that of the co-varying synoptic-scale meteorological pattern. They find that the FLC dissipation time is significantly later on high BBA days, which is mainly attributed to the longwave radiative effect of the co-transported moisture and subsequent changes in the regional circulation pattern and atmospheric heating profiles. Although the ridge regression model they trained is able to reproduce the statistical mean difference in the FLC dissipation time between high and low BBA days, it fails to provide definitive conclusions about BBA effects, due to the underfitting issue.*

*The manuscript is well written and enjoyable to read. I find this work appears of sound methodology and is of great interest to the community, with particular implications for the climate and the hydrological cycle of the Namib Desert.*

*I do have a few minor points/comments on the paper that I would like the authors to consider and address, in order to improve the manuscript's clarity and the soundness of the conclusions.*

First of all, we would like to thank the reviewer for the positive comments and for providing very helpful and constructive feedback. We thank the two reviewers in the acknowledgments of the updated version of the manuscript as follows "We are also grateful to the two anonymous reviewers for their careful and constructive feedback, which has helped improve the manuscript."
The comments and suggestions of reviewer 2 are incorporated in italics and addressed by the authors in blue.

**Comments**

**Attribution to BBA semi-direct effect**

*To me, the evidence that the authors demonstrated for a dominant role of the changing meteorological patterns (circulation, heating profile, moisture LW effect) in delaying FLC dissipation time is convincing and robust, but there are two subtleties to this that I think the authors could address/discuss in the paper.*

*The definition of 'semi-direct' effect of BBA, one thing I think this paper wasn't very clear about is the definition of the 'semi-direct' effect (appears in the title) or BBA effect, one could argue it's the local effect of BBA on FLC, all else equal, or, one could define it as the net, integrated impact of the presence of BBA on FLC (accounting for large-scale circulation adjustments). For example, Diamond et al. (2022) discusses both the "large-scale" and "local" semi-direct effects of BBA on low clouds in the SE Atlantic.*

It is true that additional precision regarding semi-direct effects definition was necessary in the manuscript.
- Added in the introduction: "Semi-direct effects in this study refer to the "large-scale" semi-direct effects, as defined in Diamond et al. (2022), involving atmospheric thermodynamic, stability, and circulation adjustments resulting from the absorption or scattering of solar radiation by aerosols."

*Changes in meteorological conditions and coastal circulation that the authors focus on discussing in the paper, to me, is part of this "large-scale" BBA/moisture semi-direct effect. However, there could be potential contributions simply from the spatiotemporally correlated synoptic patterns and regional BBA transport, which has nothing to do with the moisture LW effect or the BBA absorption, one such example is the mid-latitude intrusion events (more frequent in Sept.) that can constrain the smoke plume closer to the continent (e.g., Zhang & Zuidema, 2021, ACP). I wonder if the method used in this study can indicate/control such correlations?*

By comparing the synoptic differences on typical (i.e., averaged) high and low BBA days, the method may not capture relatively uncommon events, such as the intrusion events you mentioned. These events can significantly impact BBA transport and distribution on specific days, but they will not be clearly represented in the averaged synoptic situations used in this study. This an additional limitation that should be discussed:
- Added at the end of the Section 3.2: "However, by comparing situations averaged over hundreds of days, out-of-the-ordinary events, such as mid-latitude intrusion events (Zhang and Zuidema, 2021), which can significantly impact BBA transport and distribution on specific days, are not well captured in this study."

**Clarity on the methodology**

*It wasn't clear to me whether only June-October are used in the analyses or the whole year was used? (L82 says 15 years of data are used while L344 says only months June- October are used)*

Only June–October is selected from the 15 years of available data. We modified the sentences for better clarity.
- Modified in Section 2: "As the study focuses on the interactions between FLCs and BBA, all analyses are conducted during the BBA season from June to October (De Graaf et al., 2014) over a 15-year period (2004–2015)"
- Modified in the conclusion: "During the BBA season (June–October) of the investigated 15-year period (2004–2018)"

*L83 'some of the analyses' and L128 'For some analyses,' are not clear, please clarify and be specific*

Modified in Section 2: "...**all analyses** are conducted during the BBA season from June to October..."
Modified in Section 2.3: "**If not explicitly indicated,** 8 UTC fields are selected..." the 8 UTC fields are used every time for the ERA5 data except for Fig. 6 a) & b) where every morning time step is used.

*L103, please briefly summarize how does this work, such that one doesn't need to go to Pauli et al. (2022) to grasp the idea of this method (one can of course read it if more details are sought).*

We added information about the method.
- Modified in Section 2.1: "This method, as described by Pauli et al. (2022), employs logistic regression to predict the probability of a data point belonging to one of two binary classes: FLC or no FLC. By defining the transition between classes when this probability exceeds 50%, it becomes possible to determine the time of the transition from one class to the other and, as a result, determine the formation and dissipation times of each individual FLC event."

*What are the predictors used in the ridge regression? It says broadly "the predictors are the spatial fields of ERA5 meteorology and BCAOD from CAMS" I assume all the variables mentioned in Section 2.3 are included in the training?*

The predictors are now explicitly listed: "The predictors are the spatial fields of MSLP, Z650, T2M, SST, Q650, Q975 and EIS (as defined in Section 2.3 from ERA5 and BCAOD from CAMS."

*What exactly are the thresholds for BCAOD (25th and 75th percentile)? (a histogram as a supplementary figure would be nice)*

For the classification into the low and high BBA conditions, before using a percentile approach we tried to define common AOD thresholds across the two regions. However as the distributions are quite different between the Angolan Namib (AN) and the Central Namib (CN), it was not possible to define reliable thresholds with approximately the same number of members in both groups and both regions.
- We added a figure in Appendix A (Figure 1 in this document) in Appendix A showing the distribution of CAMS BCAOD.
- Added in Section 2.2: "In the CN, the first quartile is characterized by a high density, followed by an almost exponential decrease into and throughout the third quartile. In contrast, the AN exhibits a more linear increase in density within the first quartile, followed by a gradual decline as BCAOD increases (see Appendix A). These differences are attributed to the distance from emission sources."

[Figure]

Figure 1: BBA season mean climatology (2004–2018) of the BCAOD probability density distribution in the AN and CN, along with their respective 1st and 3rd quartiles.

*How good is this column-integrated BCAOD reanalysis, since there are aircraft measurements from the field campaigns in the region, I wonder if this product can be validated against observations?*

Although aircraft measurements are proven to be very useful, they may not be the most appropriate for evaluating the columnar AOD used in this study.
Gueymard and Yang (2020) conducted an extensive validation of CAMS using AERONET observations. The authors of this study write in their conclusion: "From the global analysis presented here, it can be concluded that, for many applications, AOD data from reanalyses such as MERRA-2 or CAMS offer significant advantages over more customary satellite observation".

*I wonder if the authors have considered using the above-cloud AOD (ACAOD, by Kerry Meyer at NASA) product to indicate BBA loading, given it's more observationally-based*

Thank you for the suggestion, this is indeed an excellent product. However, we prefer to use reanalysis data instead of satellite products or field campaigns (mentioned in the previous comment) for several reasons: there are no spatiotemporal data breaks, cloud interference is significantly reduced, it offers hourly temporal resolution, and it is convenient to combine with ERA5 data as predictors in our statistical model, requiring only an interpolation of ERA5 onto the CAMS grid.

*L120, assuming 15 years of Jun-Oct days are used, a 25th percentile will yield 562 days for each 'high BBA' and 'low BBA' group, is there additional screening involved? or missing data?*

Yes there is additional screening involved: because of the CAMS resolution, there are very few pixels per region (CN and AN). We chose a rather aggressive approach by discarding every day where at least one pixel has missing data to have a consistent number of CAMS pixels per region. This leads to approximately 300 days per quartile as written in Section 2.2.
- Added in Section 2.2 : "They are referred to as 'high BBA days' and 'low BBA days,' **each containing around 300 days after discarding those where at least one pixel of the region is outside the defined thresholds.**"

*L194, why only 200 days? instead of 300 days (L120)*

From the 300 days we get from CAMS, we discard again around 100 days with missing data in the FLS dataset (for example during clear sky days) and we end up with around 200 days per group in Fig.3 as written in Section 3.1.
- Added in Section 3.1: "Then, the data is separated into two groups of high and low BBA loading with around 200 days in each group **after discarding days without FLC events.**
- Figure 3 modified to show the number of datapoints per bar.

*In Section 2.4, CALIPSO is introduced, but I don't see how it is used. I couldn't find anywhere in Section 2 or 3 where CALIPSO data is used/discussed. L279 states the lack of skill of the statistical model could be due to the lack of vertical information of BBA (it confuses me as I thought you used CALIPSO to get the vertical dimension).*

We used CALIPSO to obtain the latitudinal and longitudinal cross-sections in Figure 2. This data is utilized to characterize the cloud and aerosol layers, as well as their relative positions. Unfortunately, it is not feasible to implement this data as

a predictor in our statistical model due to several differences between the satellite data and the reanalysis data, including missing data, differences in temporal and spatial resolution or cloud interference.
- Added in Section 2.4: "**To characterize the cloud and aerosol layers, as well as their relative positions, data** from the active-sensor platform of the Cloud-Aerosol Lidar and Infrared Pathfinder Satellite Observation (CALIPSO) is **used and presented in Figure 2.**"

**Statistical model**

*In general, I wonder what's the rationale to stick to this ridge regression linear model (given the low R2 values), over other non-linear models, such as CNN or random forest??*

A CNN or random forest may seem like a good idea, but unfortunately, neither will work for this study. For CNNs, we simply do not have enough data (around 200 days per group) to achieve proper model performance. In the case of random forests, the problem lies in the high number of correlated predictors, which provide redundant information, reducing the effectiveness of the random feature selection and increasing the risk of overfitting. We use ridge regression, which works well with our relatively small amount of data and addresses the issue of correlated predictors through regularization. Additionally, it allows us to interpret the model's predictions using the coefficient map, providing a clear understanding of the model, something that is harder to achieve with more complex models like CNNs.

*What's the ensemble spread in terms of member skill and the prediction of the mean delay in dissipation time?.*

The ensemble spread in terms of member skill is 0.04 in the AN and 0.03 in the CN
The ensemble spread in terms of the prediction of the mean delay in dissipation time is 7 minutes in the AN and 9 in the CN.
- Modified in Section 3.3: "The ensemble average skill in the AN is $R^2 = 0.34$ with a standard deviation of $0.04$, and in the CN, it is $R^2 = 0.30$ with a standard deviation of $0.03$."

*Could you show the scatter plot between truths and predictions for this model?*

We added the plot to the Appendix C (Figure 2 in this document). The spread around the y = x line is quite large, with a tendency for over-predicting in both AN and CN. Nevertheless, the highest density of points can be found around the y = x line. These results are expected given the R² values of the models.
- Added in Section 3.3: "The plot of actual versus predicted dissipation times (see Appendix C) exhibits a relatively large spread around the line of perfect agreement, with a tendency for over-prediction in both AN and CN. Nevertheless, the highest density of points broadly follows the diagonal line."

[Figure]

Figure 2: Ensemble mean predicted vs. true values for dissipation times [UTC] for the AN (a) and CN (b) models. The red dashed line represents the y = x line, indicating where predicted values equal true values.

*As mentioned in L279, have the authors tried other predictors to try to improve the skill of the model?*

Other meteorological fields were tested such as Boundary Layer Height, U and V components of winds at 650 hPa and at 10 meters or relative humidity. This didn't lead to a higher skill but increased the overfitting. The predictors are already highly correlated and therefore we argue that the model has indirect access to information not explicitly present as predictors (for example coastal circulation with MSLP). So we tried to select as few predictors as possible to reduce the risk overfitting but still have a complete picture of the meteorological situation. Nevertheless, some important information for the prediction of FLC dissipation is probably missing. One important aspect in the lifetime of FLCs is the temporal evolution of the meteorological fields, but here we only have a fixed picture at 08:00 UTC. Therefore, adding some information about the temporal variations of these fields before the dissipation could also increase the skill. Additionally, the predictand (dissipation times) is probably too complex to be accurately modeled by a linear model, but despite its limitations, this model still improves our understanding of the system.

**Editorial**

*L103, what is 'logistic regression'? a typo?*

Logistic regression is a statistical method used for binary classification tasks, where the goal is to predict the probability that a given input belongs to one of two categories.

Pauli et al. (2022) applied the logistic regression on the data binary data obtained by Andersen and Cermak (2018) to get formation and dissipation times of FLC events.
- Modified in Section 2.1: "This method, as described by Pauli et al. (2022), employs logistic regression to predict the probability of a data point belonging to one of two binary classes: FLC or no FLC. By defining the transition between classes when this probability exceeds 50%, it becomes possible to determine the time of the transition from one class to the other and, as a result, determine the formation and dissipation times of each individual FLC event."

*L133, do we know which direction?*

ERA5 simulations are steered towards a static baseline of climatological means. This baseline does not change dynamically with atmospheric conditions or real-time emissions, such as biomass burning events for example.
- Changed in Section 2.3: "Therefore, it is important to keep in mind that this prescribed forcing constrains the model's radiative environment by limiting the effects of aerosol variability."

*Figure 2, cross and star labels on the map is reversed?*

Yes thank you for pointing this error. This is now fixed.

*Figure 3, caption, please define IQR at first use.*

Caption modified in Figure 3 and 7 to: "... the whiskers extend to the highest (lowest) data points still within 1.5 times the interquartile range from the third (first) quartile.

*L230-235, what's the reason for such a strong land-sea contrast (reversed in the sign) in terms of T2M difference between high and low BBA days?*

We interpreted this as a heat low which is created by the long-wave absorption in the free troposphere caused by water vapor and BBA aerosols. Northerly advection of warm air masses is also likety to contribute to this heat low.
- Added in Section 3.2: "As such, the observed negative pressure anomalies are a clear sign of a heat low anomaly, **which is defined as an area of low atmospheric pressure caused by intense surface heating.** This phenomenon seems to be mainly driven by the greenhouse warming of the moist free-tropospheric air masses."

*L349, as mentioned in my first comment, BBA effects can include these large-scale circulation adjustments, depending how one defines it.*

Yes so not only there is the issue of disentangling water vapor effects from BBA effects but also the circulation adjustments included in the large-cale semi-direct BBA effects.
- Modified in the conclusion: "However, the grouping of high and low BBA days has been shown to lead to meteorological sampling biases, complicating the separation of meteorological effects from the **"large-scale"** semi-direct BBA effects."

*L366, perhaps, given the primary role of BBA - modulating the large-scale synoptic pattern/circulation, the use of this regression model is not suited?*

Whether it is due to missing information in the predictors, non-linearity of the system, or a predictand that is too complex, the model exhibits underfitting issues. That said, it still provides valuable insights by quantifying the processes leading to delays in the dissipation times of FLC in the region. That's why we argue that, despite its limitations, the model is well-suited for this study and improves our understanding of the system.

*L375, one possible way of doing this could be the "synoptic matching (or locking)" method used in Quaas et al. (2021, ERL), where, given a location and time, they search in their climatological database for a day with the synoptic pattern that matches the current one the best, but with different aerosol states.*

This is a nice idea; however, as we can see in this study, the synoptic patterns in the region between high and low BBA days are so different that they will likely lead to an extremely small data sample. Therefore, I don't think this approach would work here. However, expanding the study area or changing the definitions of high and low BBA days could definitely result in an interesting study.

*L375-377, given that CALIPSO is used (?) already in this study and that the key issue is the covariation between meteorology and BBA, I struggle to see how can EarthCARE offer better ways to disentangle aerosol effects.*

You are right, EarthCARE will improve our understanding of aerosl-cloud interactions but it will not directly address the covariation between meteorology and BBA.
-Deleted in the conclusion: "and potentially help disentangle aerosol effects"

**References**

Andersen, H. and Cermak, J.: First fully diurnal fog and low cloud satellite detection reveals life cycle in the Namib, Atmospheric Measurement Techniques, 11, 5461–5470, https://doi.org/10.5194/amt-11-5461-2018, 2018.

Gueymard, C. A. and Yang, D.: Worldwide validation of CAMS and MERRA-2 reanalysis aerosol optical depth products using 15 years of AERONET observations, Atmospheric Environment, 225, 117 216, https://doi.org/10.1016/j.atmosenv.2019.117216, 2020.

Pauli, E., Cermak, J., and Andersen, H.: A satellite-based climatology of fog and low stratus formation and dissipation times in central Europe, Quarterly Journal of the Royal Meteorological Society, 148, 1439–1454, https://doi.org/10.1002/qj.4272, 2022.

---

## Author Comment (AC2)

**Response to comments from reviewer 1**

**Manuscript:** A satellite-based analysis of semi-direct effects of biomass burning aerosols on fog and low cloud dissipation in the Namib Desert                                              # EGUSPHERE-2024-1627

**General comments**

*Fog is an important source of moisture in arid regions, and thus, the question of when and where fog is present is of high importance. This well-written paper investigates how biomass- burning aerosols (BBA) influence the (diurnal) dissipation time of fog/low cloud (FLC) events in Namib. Various satellite data and reanalysis data serve to analyze differences between high and low biomass burning conditions revealing significant differences also in respect to the meteorological conditions. Finally, a statistical model is built to predict the dissipation time from meteorological fields, showing the difficulty of disentangling aerosol and meteorological effects. The paper is certainly interesting for ACP and of high quality, however, I have several comments/ideas for further improvement.*

First of all, we would like to thank the reviewer for the positive comments and for providing very helpful and constructive feedback. We thank the two reviewers in the acknowledgments of the updated version of the manuscript as follows "We are also grateful to the two anonymous reviewers for their careful and constructive feedback, which has helped improve the manuscript."
The comments and suggestions from reviewer 1 are incorporated in italics and addressed by the authors in blue.

*The area of investigation is rather void of observations. Therefore, reanalysis highly depends on satellite data, which have problems in resolving the boundary layer, and the underlying NWP model. Because the 4D-Var data assimilation uses 12-hour windows (from 09 UTC to 21 UTC), jumps can occur at these times in parameters (such as water vapor) that are not constrained strongly by observations. This might influence the results in Fig. 6 as the largest changes exactly occur around the 9 UTC time. As an example, you can find the diurnal cycle of total column water vapor for Iquique, Chile, at the coast of the Atacama. Here the 21 UTC jump is most pronounced. I find the physical explanation for the behavior sound, but it would be good to check the consistency of your data and mention ERA5 limitations in the manuscript.*

Figure 3 (in this document) shows the diurnal TCWV anomalies for the Namib desert so it is possible to compare it to the data over Iquique. The jumps related to the data assimilation schemes are present, with the 21 UTC jump being the most pronounced. But the jumps over the Namib have a smaller amplitude in comparison with Iquique. Figure 4 (in this document) presents the same plot for T2M anomalies. Although jumps are still present, they are more difficult to discern compared to TCWV. Larger discontinuities probably occur at higher hPa layers. However, in Figure 6 of the manuscript, the impact of the 9 UTC jump is certainly small compared to the effect of sunrise on air temperatures.
It's a good point a should be mentioned as an ERA5 limitation:
- Added in Section 2.3: "An additional limitation of the data is related to the assimilation scheme of ERA5, which uses 12-hour windows from 09 UTC to 21 UTC (Hersbach et al., 2020). In regions with sparse observations, such as the Namib Desert, the reanalysis heavily relies on satellite data and the underlying numerical weather model, which can result in discontinuities during these times."

*I like the classification into low and high BBA conditions, which have, by definition of the percentiles, the same number of members. However, I am wondering whether one could be even more successful with an event-based approach, as I suspect that the BBA effect is most effective during selected episodes when filaments of moist and polluted air arrive in the target regions. Especially for the southern CN region satellite total column water vapor images sometimes hint at this. Therefore, it would be good to see the frequency distribution of CAMS BCAOD to illustrate how the 25 and 75 percentiles are derived and also how they differ for both regions.*

For the classification into the low and high BBA conditions, before using a percentile approach we tried to define common AOD thresholds across the two regions. However as the distributions are quite different between the Angolan Namib (AN) and the Central Namib (CN), it was not possible to define reliable thresholds with approximately the same number of members in both groups and both regions. The event-based approach is a good idea but it will lead again to issues with the number of data points as we need a relatively high and similar number of data points between the different groups to have significant results.
- We added a figure in Appendix A (Figure 1 in this document) in Appendix A showing the distribution of CAMS BCAOD.
- Added in Section 2.2: "In the CN, the first quartile is characterized by a high density, followed by an almost exponential decrease into and throughout the third quartile. In contrast, the AN exhibits a more linear increase in density within the first quartile, followed by a gradual decline as BCAOD increases (see Appendix A). These differences are attributed to the distance from emission sources."

[Figure]

Figure 1: BBA season mean climatology (2004–2018) of the BCAOD probability density distribution in the AN and CN, along with their respective 1st and 3rd quartiles.

*Coming back to my suspicion that the strong pollution mainly occurs as episodes. This counteracts the assumption of a linear model to predict FLC times. Why didn't the authors perform trajectory calculations (such as in Andersen et al. (2020) to check the meteorological similarity of the different BBA events?*

Trajectory calculations were indeed the original plan for this study, but long-term issues with large-scale data storage at our institution made it impossible to work with back trajectories. As a result, we adapted our approach and opted for a statistical framework, which has been successfully applied in this region for different predictands. Additionally, using a linear model facilitates the interpretation and visualization of coefficient fields; however, it requires a linearity assumption that is rarely fully achieved in real life, as you pointed out. In the end, due to differing synoptic conditions during high and low BBA days, similar trajectories are sparse. Therefore, statistically modeling the dynamical contribution to fog dissipation may be a more appropriate method after all.

*The target of the study is the dissipation time. Therefore, it would be good to provide more information on its variability, not only the aggregated statistics (boxplot) from Fig.3, especially as this variability is not well reproduced by the regression model (Fig.7). For these figures and all further analysis two geographical regions (AN, CN) are considered over which averages are provided. Even after looking at the study by Anderson and Cermak (2018) the motivation to put everything together in the two regions was not clear for me. Why not look at continuous development as a function of latitude and similarly as a function of distance from the coast (which the authors themselves mention to have a strong influence)? Maybe this information could be shown graphically to motivate the choice of the two regions. In this respect, it is also not clear to me how clear sky days are treated, e.g. coastal clearings.*

Working with continuous development as a function of latitude and distance to the coastline is interesting but it entails training a ridge regression for each gridpoint of the dataset. It would exponentially increase the computational time and then there is the question of how to communicate the results. It would not be possible to show the coefficients maps we produced in this study for each cell of the dataset. In this approach some form of data aggregation would still be required. By selecting these two regions (AN and CN) we focus on the two regions with the most frequent FLC events, so with the most number of data points and with contrasting characteristics in terms of aerosols: AN is closer to the emission sources.
Any absence of FLC events such as clear sky days, coastal clearings, or FLC events too brief to derive dissipation time are discarded and are treated as missing data:
- Added in Section 2.1: "Additionally, any absence of FLC events, such as during clear sky days, is discarded and treated as missing data"
- Added more information about the disisaption time in Section 2.1: "The dissipation of FLC features a main phase that begins around 6 UTC and reaches a maximum at 8 UTC in both regions, followed by a decrease until a daily minimum is observed around 13 UTC; this decline begins slightly earlier in the CN region. A secondary phase of dissipation begins at 16 UTC in the AN and at 18 UTC in the CN, continuing throughout the night, though it involves a considerably smaller number of occurrences in both regions."

*The last paragraph of the introduction contains the hypothesis states that BBA events modify the inversion and the early morning development of the planetary boundary layer. Partly, this is nicely shown in Fig. 6) but what about other parameters, such as boundary layer height or water vapor (as Q975 is so important in the ridge regression)? Vicencio et al. (2023, Fig. 10) showed that the Namib has an especially high variability in boundary layer height in austral summer. Therefore*

*Q975 might sometimes be in the BL and sometimes not.*

In Fig. 10 of Vicencio et al., the authors used daily averages for the BLH values. In this study, all meteorological fields are taken at 08 UTC, which corresponds to the most frequent dissipation time. By focusing on this specific time, we avoid a significant source of variability observed in Vicencio et al. On a typical day, we can expect the Q975 to be within the boundary layer, as the boundary layer has already started developing by this time. While exceptions are possible, they are likely to be rare.
- We added a figure in Appendix B (Figure 2 in this document) showing the hourly development of the boundary layer height in the two region on high and low BBA groups.
- Added in Section 3.2: "Additionally, the morning development of the boundary layer height (see Appendix B) indicates that the planetary boundary layer (PBL) deepens slightly more until noon on low BBA days. In the CN, the PBL is marginally lower on high BBA days; however, the differences are minimal, as shown by the largely overlapping standard deviation areas."

[Figure]

Figure 2: BBA season mean climatology (2004–2018) of the mean hourly boundary layer height in the AN **(a)** and the CN **(b)**.

**Specific Comments**

*Abstract: Mention that you are looking at the time of day when the fog is dissolving. The reader could also think that dissipation time is the duration of a fog event.*

Modified in the Abstract: "This is done by investigating both **the time of day when FLCs are dissolving** and synoptics depending on BBA loading."

*Introduction: please briefly explain the diurnal coastal circulation*

Added in the Introduction: "The diurnal coastal circulation in the region features sea breezes during the day as cooler ocean air moves inland and land breezes at night as cooler air flows from land to sea (Lindesay and Tyson, 1990)"

*L25 better represent instead of resolve*

Changed in the text.

*L74 define semi-direct*

Added in the introduction: "Semi-direct effects in this study refer to the "large-scale" semi-direct effects, as defined in Diamond et al. (2022), involving atmospheric thermodynamic and stability adjustments resulting from the absorption or scattering of solar radiation by aerosols."

*L74-79: Here, it would be good to mention briefly how you want "to disentangle the BBA effects from other meteorological covariates."*

Added : "... and used in a statistical learning framework to quantify **and disentangle** meteorological and BBA influences on FLC dissipation time."

*L97: Does 97% mean the hit rate excluding false positives? Or the CSI?*

The 97% is the Percent Correct so the correct predictions, both true positives and true negatives, to the total number of predictions made. The hit rate excluding false positives is 94% and the CSI is 83%.
- We added more information in Section 2.1: "Extensive validation against surface observations has shown a good performance (**probability of detection of 94%, a false-alarm rate of 12% and an overall correctness of classification of 97%.**)"

*Section 2: I had difficulties to extract information on data amount and resolution.*
*Amount of data**: June to October is roughly 150 days; with 15 years, this makes 2250 days. However, you have about 200 samples in one quartile, making the total sample about 800 days. What about the rest?*

*Resolution of SEVIRI: My assumption is that each area has a grid (about 6 pixels in longitudinal and many in latitudinal direction). For each gridpoint, you have (at most) one dissipation time per day. Is this correct? If yes, then it may help to delete the 15- minute resolution in line 109. It would also help to say in Fig. 3 how many data points are going into each bar. have: are available*

Amount of data: You're are right but there are two steps where we drop NaN values. Because of the CAMS resolution, there are very few pixels per region (CN and AN). We chose a rather aggressive approach by discarding every day where at least one pixel is outside the defined quartiles to have a consistent number of CAMS pixels per region. This leads to approximately 300 days per quartile as written in Section 2.2. From these 300 days we discard again around 100 days with missing data in the FLS dataset (for example during clear sky days) and we end up with around 200 days per group in Fig.3 as written in Section 3.1.
- Added in Section 2.2 : "They are referred to as 'high BBA days' and 'low BBA days,' **each containing around 300 days after discarding those where at least one pixel of the region is outside the defined thresholds.**"
- Added in Section 3.1: "Then, the data is separated into two groups of high and low BBA loading with around 200 days in each group **after discarding days without FLC events, which had no data on FLC dissipation time.**
- Figure 3 modified to show the number of datapoints per bar.

Resolution of SEVIRI: Yes exactly we have a dissipation time per gridpoint. But because of the temporal SEVIRI resolution, there are "only" 96 possible different dissipation times: from 00:00 to 23:45 with a time step of 15 minutes.
- Modified in Section 2.1: The resulting dataset provides the daily UTC time of FLC dissipation for the period of 2004–2018, **with a 15-minute temporal step (allowing for 96 possible dissipation times from 00:00 to 23:45)** and a 3x3 km spatial resolution.

*Fig.2 Why don't you show this separately for the high and low BBA situations? This would strongly help to better understand the differences between the AN and CN region.*

It's a good idea, and we considered it, but the issue is that the current plot (Fig. 2 in the manuscript) already shows some artifacts due to the spatial sparsity of the CALIPSO observations. By selecting data only from the 200 high and low BBA days, the artifacts become more pronounced, which is why we chose to present these plots. However, despite this issue, the data still reveal interesting information that should be mentioned in the paper:
- Added in Section 3.1: "In both regions, the aerosol plumes are located higher during high BBA days, around 550 hPa, whereas on low BBA days, they are situated around 750 hPa. This pattern is likely due, for the most part, to the large-scale atmospheric processes responsible for the transport of the aerosol plumes into these regions."

*L117: Please motivate why early morning. – this only comes later in L128. At this time the reader does not know that early morning is the most frequent dissipation time*

Added in Section 2.2: "Half-day averages from 00:00 to 12:00 UTC are used to capture BBA that may influence the morning dissipation of FLCs**, which is the most frequent dissipation time (Andersen and Cermak, 2018).**"

*L139: Please explicitly say in contrast to all other data another time range is used due to...-*

Added in Section 2.4: "In contrast to all other data in this study, 13 June 2006 is the earliest available date because CALIPSO was launched in April 2006."

*L170: Is the area large enough to encompass the important transport aspects? To make it easier for the reader, It would be good to mark the two regions in Fig. 4*

A tradoff is necessary when defining the spatial extent for the predictor fields: the area must be large enough to capture relevant information while remaining small enough to minimize noise and reduce the risk of overfitting. The areas defined in this study were selected after several iterations to find the best possible compromise. While they may seem small for fully encompassing all aspects potentially affecting aerosol transport, we argue that this information is indirectly represented in the contribution maps (Fig. 9) due to the separation into high and low BBA days.
We are concerned that marking the spatial extent used for the ridge regression in Fig. 4 may cause confusion, as we are not discussing or utilizing ridge regression at this point in the results section.

*L163: What is actually the resolution of your predictand. Is it the average FLC in one region per day or do you do it for each satellite pixel? I assume the average as you compare it in Fig.7 with the observed medium dissipation time? Still, this could have effects on the variability of FLC, which is poorly predicted.*

Yes, your assumption is correct; the resolution is the median FLC dissipation in one region per day. While this could have some effects on the predicted variability, these effects are likely relatively small compared to the other reasons cited in the manuscript. Because, as you mentioned, we compare it to the observed median dissipation times shown in Figure 3 of the manuscript, which, although averaged, exhibit a significantly larger spread than the predicted times, as illustrated in Figure 7.
- Added in Section 2.5: "However, the rescaling of the predictand is unnecessary because a spatial **median** of dissipation times is calculated **for each region** at each time step.

*L164: It might be lengthy, but it is good to have the list of variables of the "spatial fields of ERA5 meteorology" used as predictors. Is subsidence included?*

- Added in Section 2.5: "The predictors are the spatial fields of MSLP, Z650, T2M, SST, Q650, Q975 and EIS (as defined in Section 2.3) from ERA5 and BCAOD from CAMS"
While subsidence is not included in our analysis, we first utilized boundary layer height (BLH), which is related to the subsidence. However, the coefficient fields were quite noisy, making it difficult to extract useful information. Although subsidence might yield better results than BLH, we do not expect significant differences in the model's performance by including it.

*L198: Mention the definition of variability, IQR?*

Added in Section 3.1: "While the variability of dissipation times, **measured by the interquartile range**, does not change with BBA in the AN"

*L206: Motivate 650hPa. It is done later but before wrote that the transport is highest in 650 hPa*

- Moved: "the typical altitude of the BBA plumes" from line 212 to the beginning of Section 3.2
In Section 3.1 when describing Figure 1, we now provide a range of pressure layers where the BBA plumes are located. However, 650 hPa is the midpoint, where BBA plumes are most likely to be found.

*L247: grid cell is better than pixel here*

Changed in the text.

*L183. Z and Q are not defined.*
*L231: T2M not defined*

All the ERA5 acronyms are defined in the Section 2.3. But we also chose to not use the acronyms in the captions of the figures so a reader could understand the parameters of the figures without needing the next.

*L275: A correlation coefficient of 0.58 (explained variance 0.34) is not too bad. Did you try to identify the cases when especially poor predictions were made, e.g. by colouring scatter diagrams etc?*

It's a good question. Examining the extreme values reveals a relatively big spread. For AN the worst model has a $R^2$ of 0.22, while the best model has a $R^2$ of 0.41 (and a similar spread for CN). This can largely be attributed to the choice of using a random split for the training and test groups. In some cases, models may encounter days with vastly different synoptic situations in each group, causing them to fail to generalize and resulting in low $R^2$ values. Conversely, random splits can also lead to temporal autocorrelation when neighbouring days appear in both the training and test groups, which may affect the robustness of the model. By creating an ensemble mean of 50 different models, we effectively average out these issues. For comparison, we ran a single model using time series cross-validation, thus avoiding temporal autocorrelation, and found similar $R^2$ values for both regions, with nearly identical patterns in the coefficient fields.

*L323:"These contributions are due to a negative SST anomaly close to the coastline which may be explained by the greenhouse warming in the free troposphere". Not clear to me..*

Modified to "These contributions are the product of a negative coefficient field and a negative SST anomaly near the coastline (not shown), which could partly be explained by reduced incoming solar radiation due to absorption in the free troposphere. However, since the ocean surface responds only slowly, other factors, such as the changed circulation possibly contributes as well.

*Conclusions: Wouldn't radiative sensitivity studies also be helpful in disentangling water vapor and BBA contributions?*

That's a good idea yes, but it would probably be more oriented towards direct aerosol effects rather than semi-direct.
- Added in the conclusion: "Additionally, radiative sensitivity studies such as Obregón et al. (2018), could also be useful in

disentangling aerosol direct effects from meteorological covariates."

**Technical corrections**

*L 160: Greek letter for lambda.*

Changed in the text.

*I find that the readability of the text is degrading when figure caption information (that is not important for following the text) is repeated in the test. My recommendation would be to remove -statements such as in L210 (Additionally, in the Q975 fields (Fig. 5c and d) the subsurface regions at this pressure level are masked). Or statements such as like "are shown in the third panel.", "dashed lines"*

The highlighted sentences were deleted, as well as similar ones throughout the manuscript.

*L330. Blank missing*

Fixed in the text.

[Figure]

Figure 3: ERA5 TCWV anomaly (hourly average minus daily mean) from 2004-2018 in the Namib desert.

[Figure]

Figure 4: ERA5 T2M anomaly (hourly average minus daily mean) from 2004-2018 in the Namib desert.

---

## Author Response (AR2)

**Response to comments from editor**

**Manuscript:** A satellite-based analysis of semi-direct effects of biomass burning aerosols on fog and low cloud dissipation in the Namib Desert                                          # EGUSPHERE-2024-1627

We would like to thank the editor for the helpful and constructive feedback. The comments and suggestions are incorporated in italics and addressed by the authors in blue. We thank the editor in the acknowledgements of the updated version of the manuscript as follows: "We are also grateful to the editor, Franziska Aemisegger, and the two anonymous reviewers for their careful and constructive feedback, which has helped improve the manuscript."

*1) L.3 ", and sometimes are reaching the coastal fog and low clouds" -can you be more precise? e.g. "Each year between June and October, in some synoptic settings, absorbing biomass burning aerosols (BBA) are overlying the stratocumulus clouds in the adjacent Southeast Atlantic, thereby modulating the time of the day when FLC dissolve. In favorable synoptic conditions, this layer of BBA reaches Namibia and its desert, where it interacts with coastal fog and low clouds ."*
*This would help a bit because up until here a non-expert doesn't know what the impact of BBA is on the cloud layer.?*

- Modified to: "Each year between June and October absorbing biomass burning aerosols (BBA) are overlying the stratocumulus clouds in the adjacent Southeast Atlantic. In some synoptic settings, this layer of BBA reaches Namibia and its desert, where it interacts with coastal fog and low clouds (FLCs)"
Thank you for the suggestion, it is true that splitting the sentence in two improves clarity. Additionally, we made minor corrections, as the BBA layers can generally be considered present over the Southeast Atlantic during these months, but they only reach the coastal regions in certain synoptic settings. Nonetheless, at this point in the abstract, it seems too speculative to claim that the BBA layers impact FLC dissipation, as this is explored throughout the study. This is why we prefer to simply state that interactions are present without assuming their effects.

*2) I do agree with Rev. 1 about their point on the artificial effects due to the 12h satellite data assimilation on water vapour and TCW, you may therefore add their suggested reference at L. 150. Vicencio, J., C. Böhm, J.H. Schween, U. Löhnert, and S. Crewell, 2023: A comparative study of the atmospheric water vapor in the Atacama and Namib Desert, Global and Planetary Change, 104320, https://doi.org/10.1016/j.gloplacha.2023.104320.*
*For me personally this reference would help to understand what the implications of only assimilating satellite obs in the 12h time window are at this specific location of the text, while Hersbach et al. 2020 is a more general reference for the data assimilation window.*

We, of course, would prefer to cite this study instead of the more general reference to the data assimilation window in Hersbach et al. (2020). However, after careful examination, we did not find any mention of this limitation in the study. The figure of hourly TCWV anomalies provided by Reviewer 1 is also not present in the study. As we understand it, the reference was given in relation to Reviewer 1's fifth comment, where the variability of boundary layer height is questioned.

*3) I do also agree with Rev. 1 on the transient nature of BBA influence: "Coming back to my suspicion that the strong pollution mainly occurs as episodes. This counteracts the assumption of a linear model to predict FLC times." Can you mention this specific limitation of the chosen prediction framework more explicitly in the conclusions. Also the final sentence from the Abstract sounds a bit like a dead end. Can you add a sentence that would show the way forward? LES modelling? Use of trajectories or combination with other tracers?.*

- Added in the conclusions: "Additionally, using this model requires the assumption of linearity in the underlying processes."

- Added in the abstract: "... and invite detailed modeling analyses of the underlying processes, for example, with large eddy simulations."

*4) Semi-direct (both reviewers): is now defined in the intro but in the abstract I also stumble over it and would very much benefit from a short explanation. Either avoid semi-direct and directly say that you mean impact on thermodynamics and stability through interaction with radiation or, explain what semi-direct is in a relative clause: "In this study, a novel 15-year data set of geostationary satellite observations in the Namib Desert is used together with reanalysis data in order to better understand the semi-direct impacts of BBA on the dissipation of FLCs in the Namib desert, i.e. through adjustments of the atmospheric stability by the interaction of the aerosols with radiation.*

- Modified in the abstract: "... i.e., through adjustments of atmospheric stability and thermodynamics via the interaction of aerosols with radiation"
Thank you for the suggestion. We think that it is important for "semi-direct" to appear in the abstract but it is true that it requires an explanation.

*5) L. 78 and 80: I agree with reviewer 1 that it is unclear at this stage (and later on) how you can disentangle the BBA effects and meteorology*

- Modified in the introduction: "The goals of this study are thus to better understand possible BBA semi-direct effects on the dissipation of FLCs in the Namib and to **attempt to** disentangle the BBA effects from other meteorological covariates. To this end, a 15-year time series of geostationary satellite observations of FLCs in the Namib is analyzed together with reanalysis data to characterize situations under contrasting BBA loading, and used in a statistical learning framework to quantify and **partially** disentangle meteorological and BBA influences on FLC dissipation time."
We added these two terms to indicate that it is not fully achieved. Nevertheless, thanks to the statistical model, the BBA effects are disentangled from the coastal circulation. However, the main challenge of separating water vapor effects from BBA effects obviously remains.

*6) L. 118-121: please shortly mention why this is the most frequent dissipation time.*

- Modified in section 2.1: "In both regions, dissipation begins shortly after sunrise, reaching a maximum at 8 UTC, which can be attributed to stronger solar irradiance (Andersen and Cermak, 2018). This is followed by a decrease until a daily minimum is observed around 12 UTC."
We added an explanation of why 8 UTC is the most frequent dissipation time. Additionally, we removed the sentences about the second phase of dissipation that occurs during the night, as this phase is not relevant to this study; the semi-direct BBA effects we focus on require solar radiation.

*7) L. 300-301: Are these transient events not well captured or not captured at all? And what do you suggest as a way forward to address this point in the future?*

- Modified in section 3.2: "However, by comparing situations averaged over hundreds of days, this study does not **effectively** capture out-of-the-ordinary events, such as mid-latitude intrusion events (Zhang and Zuidema, 2021), which can significantly impact BBA transport and distribution on specific days. **While detailed case studies using back trajectories could analyze the variability of such events, they are beyond the scope of this study**"
We argue that while our averaging captures some information related to these transient events, it is not sufficient to fully discern their effects. Additionally, it is uncertain whether these kinds of events would drastically modify the synoptic situations described in the study. For example, even if the direction from which BBA are coming changes, we can still expect longwave heating to modulate the coastal circulation and impact fog dissipation.

*8) Rev.2 "How good is this column-integrated BCAOD reanalysis, since there are aircraft measurements from the field campaigns in the region, I wonder if this product can be validated against observations?" and the next remark by Rev.2 about ACAOD from NASA: I understand the motivation for using BCAOD reanalyses data in your study, but this choice should be justified explicitly in Section 2.2. Mentioning that other more observationally-based products can be used in particular for case studies and to evaluated the reanalysis product in more detail would be helpful for the reader.*

- Added in section 2.2: "While more observational-based products exist, such as aircraft measurements from field campaigns in the region, which are very useful in case studies, a reanalysis data set is preferred in this study to avoid issues related to missing data, cloud interference, and to facilitate combination with meteorological reanalysis data in a statistical framework. Additionally, the AOD data was extensively validated using observations in Gueymard and Yang (2020), where the authors found a small but existing tendency in CAMS to underestimate AOD across Africa. However, they conclude that for many applications, this data set offers significant advantages over customary observational-based products."

*9) Rev. 2's point about why not using an approach that would accommodate for the non-linear nature of the prediction problem. Please shortly explain the rationale behind using a ridge regression model at the beginning of Section 2.5.*

- Added in section 2.5: "Spatial neural networks would be an ideal tool for this; however, they require a large number of data points, and their interpretation and sensitivity estimation are more challenging. Therefore, ridge regression, a regularized linear model..."

***Technical comment:*** *°S and °N should not be in italics.*

Corrected in the text.

**References**

Andersen, H. and Cermak, J.: First fully diurnal fog and low cloud satellite detection reveals life cycle in the Namib, Atmospheric Measurement Techniques, 11, 5461–5470, https://doi.org/10.5194/amt-11-5461-2018, 2018.

Gueymard, C. A. and Yang, D.: Worldwide validation of CAMS and MERRA-2 reanalysis aerosol optical depth products using 15 years of AERONET observations, Atmospheric Environment, 225, 117 216, https://doi.org/10.1016/j.atmosenv.2019.117216, 2020.

Zhang, J. and Zuidema, P.: Sunlight-absorbing aerosol amplifies the seasonal cycle in low-cloud fraction over the southeast Atlantic, Atmospheric Chemistry and Physics, 21, 11 179–11 199, https://doi.org/10.5194/acp-21-11179-2021, 2021.